**Brief Communication**

# microbeMASST: a taxonomically informed mass spectrometry search tool for microbial metabolomics data

**A list of authors and their affiliations appears at the end of the paper**

microbeMASST, a taxonomically informed mass spectrometry (MS) search tool, tackles limited microbial metabolite annotation in untargeted metabolomics experiments. Leveraging a curated database of >60,000 microbial monocultures, users can search known and unknown MS/MS spectra and link them to their respective microbial producers via MS/MS fragmentation patterns. Identification of microbe-derived metabolites and relative producers without a priori knowledge will vastly enhance the understanding of microorganisms' role in ecology and human health.

Microorganisms drive the global carbon cycle[1] and can establish symbiotic relationships with host organisms, influencing their health, aging and behaviour[2–6]. Microbial populations interact with different ecosystems through the alteration of available metabolite pools and the production of specialized small molecules[7,8]. The vast genetic potential of these communities is exemplified by human-associated microorganisms, which encode ~100 times more genes than the human genome[9,10]. However, this metabolic potential remains unreflected in modern untargeted metabolomics experiments, where typically <1% of the annotated molecules can be classified as microbial. This problem particularly affects mass spectrometry (MS)-based untargeted metabolomics, a common technique to investigate molecules produced or modified by microorganisms[11], which famously struggles with spectral annotation of complex biological samples. This is because most spectral reference libraries are biased towards commercially available or otherwise accessible standards of primary metabolites, drugs or industrial chemicals. Even when metabolites are annotated, extensive literature searches are required to understand whether these molecules have microbial origins and to identify the respective microbial producers. Public databases, such as KEGG[12], MiMeDB[13], NPAtlas[14] and LOTUS[15], can assist in this interpretation, but they are mostly limited to well-established, largely genome-inferred metabolic models or to fully characterized and published molecular structures. In addition, while targeted metabolomics efforts aimed at interrogating the gut microbiome mechanistically have been developed[16], these focus only on relatively few commercially available microbial molecules. Hence, the majority of the microbial chemical space remains unknown despite the continuous expansion of MS reference libraries. To fill this gap, we have developed microbeMASST (https://masst.gnps2.org/microbemasst/), a search tool that leverages public MS repository data to identify the microbial origin of known and unknown metabolites and map them to their microbial producers.

microbeMASST is a community-sourced tool that works within the GNPS ecosystem[17]. Users can search tandem MS (MS/MS) spectra obtained from their experiments against the GNPS/MassIVE repository and retrieve matching samples exclusively acquired from extracts of bacterial, fungal or archaeal monocultures. No other available resource or tool allows linking uncharacterized MS/MS spectra to characterized microorganisms. The microbeMASST reference database of microbial monocultures has been generated through years of community contributions and metadata curation, and it contains microorganisms isolated from plants, soils, oceans, lakes, fish, terrestrial animals and humans (Fig. 1a). All available microorganisms have been categorized according to the NCBI taxonomy[18] at different taxonomic resolutions (that is, species, genus, family and so on) or mapped to the closest taxonomically accurate level, if no NCBI ID was available at the time of database creation. As of September 2023, microbeMASST includes 60,781 liquid chromatography (LC)–MS/MS files comprising >100 million MS/MS spectra mapped to 541 strains, 1,336 species, 539 genera, 264 families, 109 orders, 41 classes and 16 phyla from the three domains of life: Bacteria, Archaea and Eukaryota (Fig. 1b). Different from MASST[19], which uses a precomputed network of ~110 million MS/MS spectra to enable spectral searching, microbeMASST is based on the recently introduced Fast Search Tool (https://fasst.gnps2.org/fastsearch/)[20]. This tool, originally designed for proteomics, drastically improves search speed by several orders of magnitude by indexing all the MS/MS spectra present in GNPS/MassIVE and restricting the search space to the user input parameters. Because of this, search results are returned within seconds

✉e-mail: pdorrestein@health.ucsd.edu

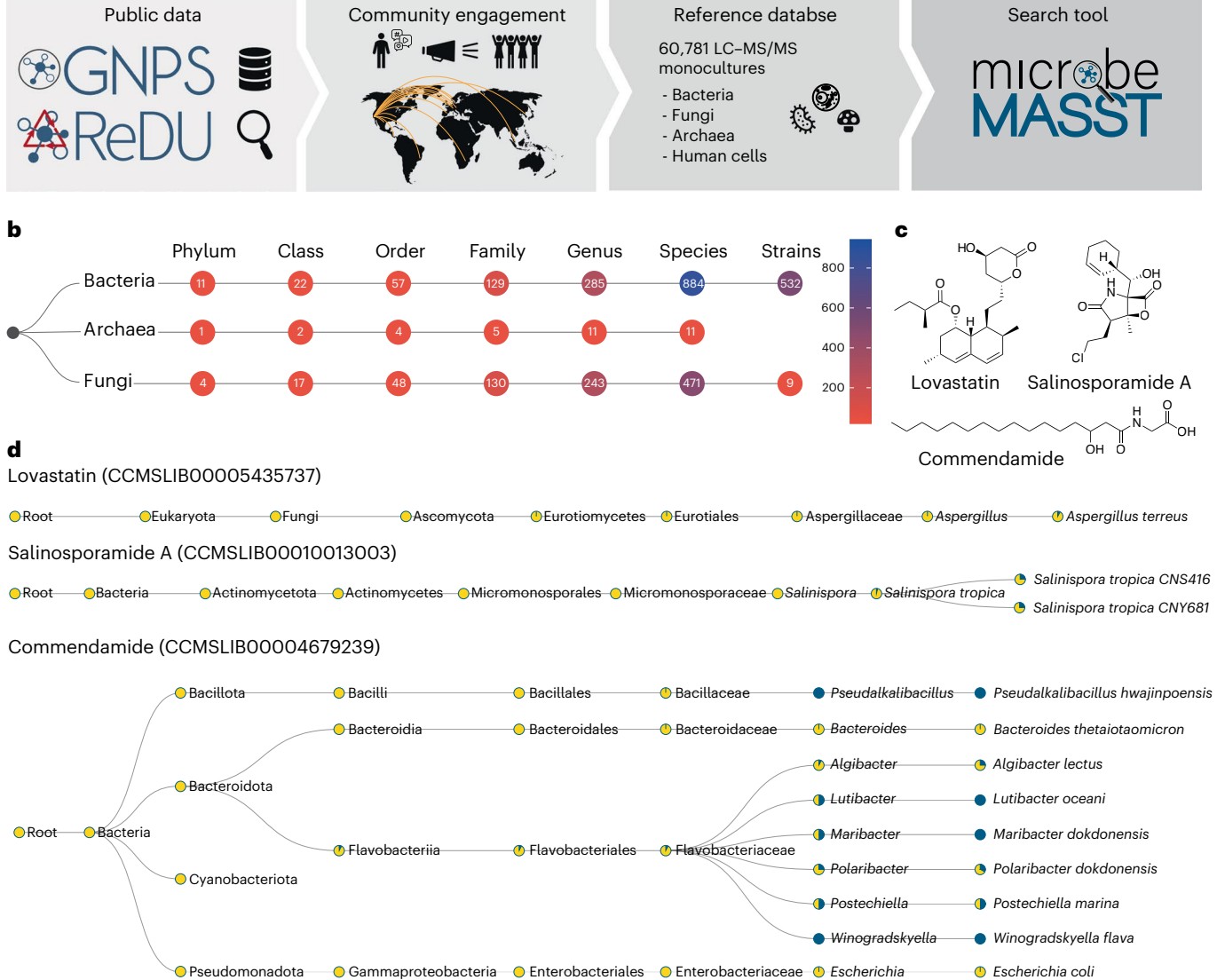

**Fig. 1 | The microbeMASST search tool and reference database. a**, Community contributions of data and knowledge to GNPS[17], ReDU[57] and MassIVE from 2014 to 2022 were used to generate the microbeMASST reference database. In addition, a public invitation to deposit data in June 2022 resulted in the further deposition of LC–MS/MS files from 25 different laboratories from 15 different countries across the globe, leading to the curation of a total of 60,781 LC–MS/MS files of microbial monoculture extracts. **b**, microbeMASST comprises 1,858 unique lineages across three different domains of life mapped to 541 unique strains, 1,336 species, 539 genera, 264 families, 109 orders, 41 classes and 16 phyla. **c**, Examples of medically relevant small molecules known to be produced by bacteria or fungi. Lovastatin, a cholesterol-lowering drug originally isolated from *Aspergillus*

genus[25]; salinosporamide A, a Phase III candidate to treat glioblastoma produced by *Salinispora tropica*[27]; and commendamide, a human G-protein-coupled receptor agonist[28]. **d**, microbeMASST search outputs of the three different molecules of interest confirm that they were exclusively found in monocultures of the only known producers. Pie charts display the proportion of MS/MS matches found in the deposited reference database. Blue indicates a match with a monoculture, while yellow represents a non-match. Searches were performed using MS/MS spectra deposited in the GNPS reference library: lovastatin (CCMSLIB00005435737), salinosporamide A (CCMSLIB00010013003) and commendamide (CCMSLIB00004679239). GNPS, ReDU and microbeMASST logos reproduced under a Creative Commons license CC BY 4.0.

as opposed to 20 min per search or 24–48 h for modification tolerant searches in the original implementation of MASST. In addition, microbeMASST leverages pre-curated file-associated metadata to aggregate results into easy-to-interpret taxonomic trees. This represents a major enhancement over MASST, where users have to manually inspect results tables and contextualize them, making interpretations tedious. Finally, users can leverage microbeMASST Python code to perform batch searches of thousands of MS/MS spectra by providing either a formatted MS/MS file (.mgf) or a list of Universal Spectrum Identifiers (USIs)[21], which represent paths to spectra in public datasets[22]. This is particularly useful for creating integrated data analysis

pipelines using the standard outputs (.mgf) of already established data processing tools, such as MZmine[23].

In the microbeMASST web app (https://masst.gnps2.org/microbe-masst/), users can search single MS/MS spectra and obtain matching results from the reference database of microbeMASST, providing either a USI or a precursor ion mass and its spectral fragmentation pattern (Supplementary Fig. 1). Analogue search can also be enabled to discover molecules related to the MS/MS spectrum of interest across the taxonomic tree[17,19,24]. The microbeMASST web app displays query results in interactive taxonomic trees, which can be downloaded as HTML files. Nodes in the trees represent specific taxa and display rich information,

such as taxon scientific name, NCBI taxonomic ID, number of deposited sample data files, number of sample data files containing a match to the queried spectrum, within the user search criteria, and a proportion of the number of sample data files matching the queried spectrum to the number of total available sample data files for that specific taxon in the reference database of microbeMASST. This proportion is also visualized through pie charts. Information for an MS/MS match in a particular taxon is propagated upstream through its lineage. The reactive interface of microbeMASST enables filtering of the tree to specific taxonomic levels or to a minimum number of matches observed per taxon. In addition, three data tables are generated, linking the search job to other resources in the GNPS/MassIVE ecosystem. For example, each MS/MS query is also searched against the public MS/MS reference library of GNPS (587,213 MS/MS spectra, September 2023) to provide spectra annotations when available. The annotations to reference compounds are listed under the 'Library matches' tab (Supplementary Fig. 2a). The 'Datasets matches' tab contains information on the matching scans, displaying scientific name, NCBI taxonomic ID and taxonomic rank, number of matching fragment ions and modified cosine score together with a link to a mirror plot visualization (Supplementary Fig. 2b). Finally, the 'Taxa matches' tab informs on how many matches were found per taxon and the number of samples available for that taxon (Supplementary Fig. 2c). Quality controls (QCs) and blank samples ($n = 2,902$) present in the reference datasets of microbeMASST have been retained to provide information on possible contaminants and media components. In addition, data from human cell line cultures ($n = 1,199$) have been included to enable assessment of whether molecules can be produced by both human hosts and microorganisms. It is important to point out that microbeMASST allows linking of both partly annotated, through MS/MS match to reference library spectra, and fully uncharacterized spectra to possible microbial producers but that technical limitations inherent to mass spectrometry or the experiment itself are present. For example, the absence of a matching spectrum in a specific taxon does not necessarily indicate that it is not capable of producing the searched molecule but rather that the methodology used to acquire the data did not allow its detection. These and other limitations are described in Methods. Despite these limitations, microbeMASST can uniquely enable the discovery of links between uncharacterized MS/MS spectra and defined microorganisms, providing valuable information for future mechanistic studies.

Search results for lovastatin, salinosporamide A and commendamide MS/MS spectra highlight how microbeMASST can correctly connect microbial molecules to their known producers (Fig. 1c). In the case of lovastatin, a clinically used cholesterol-lowering drug originally isolated from *Aspergillus terreus*[25], spectral matches were unique to the genus *Aspergillus* (Fig. 1d). The MS/MS spectrum for salinosporamide A, a Phase III candidate to treat glioblastoma[26], only matched two strains of *Salinispora tropica* (Fig. 1d), the only known producer[27]. Commendamide, first observed in cultures of *Bacteroides vulgatus* (recently reclassified as *Phocaeicola vulgatus*), is a G-protein-coupled receptor agonist[28]. Surprisingly it had many matches to several bacterial cultures, including in Flavobacteriaceae (*Algibacter*, *Lutibacter*, *Maribacter*, *Polaribacter*, *Postechiella* and *Winogradskyella*) and *Bacteroides* cultures (Fig. 1d). Additional examples include searches of mevastatin, arylomycin A4, yersiniabactin, promicroferrioxamine, and the microbial bile acid conjugates[29–31] glutamate-cholic acid (Glu-CA) and glutamate-deoxycholic acid (Glu-DCA) (Supplementary Fig. 3). Mevastatin, another cholesterol-lowering drug originally isolated from *Penicillium citrinum*[32], was only found in samples classified as fungi. The antibiotic arylomycin A4 was observed in different *Streptomyces* species, and it was originally isolated from *Streptomyces* sp. Tue 6075 in 2002[33]. Yersiniabactin, a siderophore originally isolated from *Yersinia pestis*[34] whose monoculture is not yet present in the reference database of microbeMASST, was observed in *Escherichia coli* and *Klebsiella* species, consistent with previous observations[35,36].

Promicroferrioxamine, another siderophore, was observed to match *Micromonospora chokoriensis* and *Streptomyces* species. This molecule was originally isolated from an uncharacterized Promicromonosporaceae isolate[37]. The MS/MS spectrum of the gut microbiota-derived Glu-CA, an amidated tri-hydroxylated bile acid, was most frequently observed in cultures of *Bifidobacterium* species, while Glu-DCA was found only in one *Bifidobacterium* strain but also in two *Enterococcus* and *Clostridium* species. None of the molecules were found in cultured human cell lines, highlighting the ability of microbeMASST to distinguish MS/MS spectra of molecules that can be exclusively produced by either bacteria or fungi. It is important to acknowledge that MS/MS data generally do not differentiate stereoisomers, but it can nevertheless provide crucial information on molecular families.

microbeMASST can be also used to extract microbial information from mass spectrometry-based metabolomics studies without any a priori knowledge. To illustrate this, we reprocessed an untargeted metabolomics study with data acquired from 29 different organs and biofluids comprising tissues including brain, heart, liver, blood and stool of germ-free (GF) mice and mice harbouring microbial communities, also known as specific pathogen-free (SPF) mice[30] (Fig. 2a). We extracted 10,047 consensus MS/MS spectra uniquely present in SPF mice and queried them with microbeMASST. A total of 3,262 MS/MS spectra were found to have a microbial match to the microbeMASST reference database. Of these, 837 were also found in human cell lines and for this reason were removed from further analysis. Among the remaining 2,425 MS/MS spectra, 1,673 were exclusively found in bacteria, 95 in fungi and 657 in both (Supplementary Fig. 4). These MS/MS spectra were then processed with SIRIUS[38] and CANOPUS[39] to tentatively annotate the metabolites and identify their chemical classes. A file containing all these spectra of interest can be explored and downloaded in .mgf format from GNPS (see Methods). To further validate the microbial origin of these MS/MS spectra, we assessed their overlap with data acquired from a different study comparing SPF mice treated with a cocktail of antibiotics to untreated controls[40]. Interestingly, 621 MS/MS spectra were also found in this second dataset and 512 were only present in animals not treated with antibiotics (Fig. 2b). The distribution of these spectra and their putative classes across bacterial phyla was visualized using an UpSet plot[41] (Fig. 2c). Notably, most of the spectra classified as terpenoids were commonly observed across phyla, while amino acids and peptides appeared to be more phylum specific. Of these 512 spectra, 23% had a level 2 putative annotation according to the 2007 Metabolomics Standards Initiative[42], matching the GNPS reference library (Supplementary Table 1). A level 2 annotation within the user-specified search criteria might result in MS/MS matches between molecules belonging to related families as opposed to unique molecules. Annotations included the recently described amidated microbial bile acids[19,29–31,43–48], free bile acids originating from the hydrolysis of host-derived taurine bile acid conjugates[49], keto bile acids formed via microbial oxidation of alcohols[30], N-acyl-lipids belonging to a similar class of metabolites as commendamide[28] (a microbial N-acyl lipid), di- and tri- peptides seen in microbial digestion of proteins[50], and soyasapogenol, a by-product of the microbial digestion of complex saccharides from dietary soyasaponins[30]. Part of the remaining unannotated spectra can be identified as chemical modifications of the above annotated microbial metabolites through spectral similarity obtained from molecular networking (Supplementary Fig. 5). This list of annotated MS/MS spectra included metabolites that are not yet widely considered to be of microbial origins, such as the di- and tri-hydroxylated bile acids and the glycine-conjugated bile acids[43]. One interpretation of these findings is that microorganisms are capable of producing metabolites previously described to be only of mammalian origins. Notable examples of metabolites that have been established to be produced by both the mammalian host and bacteria include serotonin[51], γ-aminobutyric acid (GABA)[52] and most recently, glycocholic acid[43,53–55]. In addition, an alternative hypothesis is that

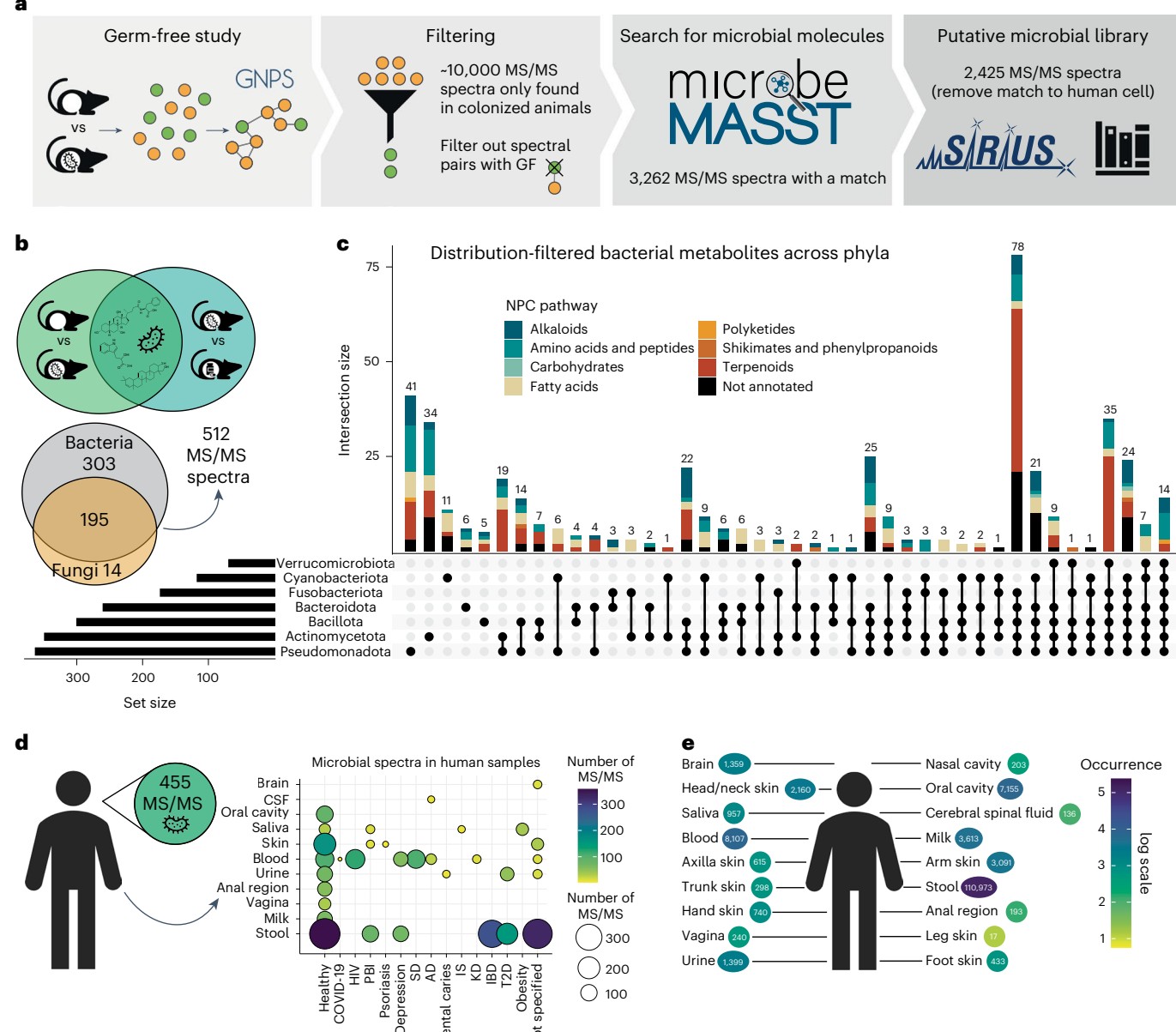

**Fig. 2 | microbeMASST can identify microbial MS/MS spectra within mouse and human datasets. a**, Workflow to extract microbial MS/MS spectra from biochemical profiles of 29 different tissues and biofluids of SPF mice that are not observed in GF mice[30]. The MS/MS spectra unique to SPF mice (10,047) were searched with microbeMASST. A total of 3,262 MS/MS spectra had a match; those MS/MS also matching human cell lines were removed, leaving a total of 2,425 putative microbial MS/MS spectra (see Methods to download .mgf file). **b**, The presence of the 2,425 MS/MS spectra was evaluated in an additional animal study looking at antibiotic usage[40]. A total of 512 MS/MS spectra, out of the 621 overlapping, were exclusively found in animals not receiving antibiotics. **c**, UpSet plot of the distribution of the detected MS/MS spectra (512) across bacterial phyla. Terpenoids were more commonly observed across phyla, while amino

acids and peptides appeared to be more phylum specific. **d**, The 512 MS/MS spectra were searched in human datasets and 455 were found to have a match. These MS/MS spectra were present in both healthy individuals and individuals affected by different diseases. **e**, Most of the MS/MS spectra (n = 411) matched faecal samples (n = 110,973 matches), followed by blood, oral cavity, breast milk, urine and several other organs. CSF, cerebral spinal fluid; COVID-19, coronavirus disease 2019; HIV, human immunodeficiency virus; PBI, primary bacterial infectious disease; SD, sleep disorder; AD, Alzheimer's disease; IS, ischaemic stroke; KD, Kawasaki disease; IBD, inflammatory bowel disease; T2D, type II diabetes. GNPS and and microbeMASST logos reproduced under a Creative Commons license CC BY 4.0; SIRIUS logo reproduced under a Creative Commons license CC BY 4.0-ND.

microorganisms can also selectively stimulate the production of host metabolites. Other limitations regarding annotations are discussed in Methods. To assess whether the observations from the mouse models translate to humans, we searched and found that 455 out of the 512 MS/MS spectra of interest matched public human data (Fig. 2d). Interestingly, these spectra were found in both healthy individuals and individuals affected by different diseases, including type II diabetes,

inflammatory bowel disease, Alzheimer's disease and other conditions. These spectra were most commonly found in stool samples (n = 110,973 MS/MS matches), followed by blood, breast milk and the oral cavity, as well as other organs including the brain, skin, vagina and biofluids (for example, cerebrospinal fluid and urine) (Fig. 2e). These findings support the concept that a substantial number of microbial metabolites reach and influence distant organs in the human body[56].

We anticipate that microbeMASST will be a key resource to enhance understanding of the role of microbial metabolites across a wide range of ecosystems, including oceans, plants, soils, insects, animals and humans. This expanding resource will enable the scientific community to gain valuable taxonomic and functional insights into diverse microbial populations. The mass spectrometry community will play a key role in the evolution of this tool in the future through the continued deposition of data associated with microbial monocultures and the expansion of spectral reference libraries. Moreover, microbeMASST holds potential for various applications ranging from aquaculture and agriculture to biotechnology and the study of microbe-mediated human health conditions. By harnessing the power of public data, we can unlock opportunities for advancements in multiple fields and deepen our understanding of the intricate relationships between microorganisms and their ecosystems.

## Methods

### Data collection and harmonization

Data deposited in GNPS/MassIVE were investigated manually and systematically using ReDU[57] (https://redu.ucsd.edu/) to extract all publicly available MS/MS files (.mzML or .mzXML formats) acquired from monocultures of bacteria, fungi, archaea and human cell lines. Only monocultures were included in the reference database of this search tool to unequivocally associate the production of the detected metabolites to each specific taxon. A total of 60,781 files from 537 different GNPS/MassIVE datasets were selected to be used as the reference database of microbeMASST (Supplementary Table 2). These include files deposited in response to our call to the scientific community. Between May and July 2022, 25 different research groups deposited 65 distinct datasets in GNPS/MassIVE, comprising a total of 3,142 unique LC–MS/MS files. This represented a 5.45% increase in publicly available MS/MS data acquired from monocultures in just 2 months. To qualify as a contributor and be credited as one of the authors, researchers had to deposit high-resolution LC–MS/MS data acquired either in positive or negative ionization modes from monocultures of either bacteria, fungi or achaea. Harmonization of the acquired data and metadata represented a challenge. The NCBI taxonomic database is constantly expanding and evolving, and the ReDU latest update (December 2021) does not accommodate the latest deposited taxa. For this reason, an additional metadata file (microbeMASST_metadata_massiveID) was generated specifically for the microbeMASST project and uploaded to the respective GNPS/MassIVE datasets deposited by the collaborators if the ReDU workflow failed. All the collected information was finally aggregated in a single .csv file (microbe_masst_table.csv) that can be found on GitHub, which contains: (1) full MassIVE path of each sample, (2) file name of each sample reported as its MassIVE ID/file name to avoid the presence of duplicated names, (3) MassIVE ID, (4) taxonomic name of the isolate as reported by the author submitting the associated metadata, (5) alternative taxonomic name if the provided taxonomic name was incorrect or not present in NCBI, (6) associated NCBI ID to the taxonomic name or the alternative taxonomic name, when present, (7) definition if the taxonomic ID was automatically assigned or manually curated, and information if (8) ReDU metadata are available for that specific file and if the file correspond to a (9) blank or (10) QC rather than a unique biological sample.

Unique taxonomic names and NCBI IDs were extracted from the metadata associated with the selected samples. When metadata were not available and multiple species of bacteria or fungi were present in the same dataset, samples were generically classified as bacteria or fungi. Concordance between taxonomic names and NCBI IDs was checked by blasting taxonomic names against NCBI (https://www.ncbi.nlm.nih.gov/Taxonomy/TaxIdentifier/tax_identifier.cgi) to obtain respective NCBI IDs and updated taxonomic names. Results were manually investigated and missing IDs were recovered using the NCBI browser (https://www.ncbi.nlm.nih.gov/Taxonomy/Browser/wwwtax.cgi). If the taxonomic name was not found in NCBI, most probably because it was not deposited yet, the NCBI of the closest taxon was retrieved and used. For example, the strain *Staphylococcus aureus* CM05 was unavailable in NCBI and was curated to the species *Staphylococcus aureus* instead.

### Taxonomic tree generation

The microbeMASST taxonomic tree was generated using both R 4.2.2 and Python 3.10. In R, the microbeMASST table was filtered and only unique NCBI IDs were retained ($n = 1,834$). The classification function of the 'taxize' package (v.0.9.100) was used to retrieve the full lineage of each NCBI ID[58]. Main taxonomic ranks (kingdom to strain) plus subgenus, subspecies and varieties were kept to obtain taxonomic trees with a similar number of nodes per lineage. The list of NCBI IDs of all lineages was then imported to Python, where the ETE3 toolkit was used to generate a taxonomic tree on the basis of the provided NCBI IDs[59]. The generated Newick tree was then converted into JSON format and information such as taxonomic rank and number of available samples per taxon was added. In addition, children nodes for blanks and QCs were created to be visualized in the same tree.

### MASST query

The microbeMASST web application was built using Dash and Flask open-source libraries for Python (https://github.com/mwang87/GNPS_MASST/blob/master/dash_microbemasst.py). The web app can receive as inputs either a USI or an MS/MS spectrum (fragment ions and their intensities). In addition, batch searches can be performed using a customizable Python script that can read either a .tsv file containing a list of USIs or a single .mgf file (https://github.com/robinschmid/microbe_masst). Through the manuscript, we showcase how we were able to search for more than 10,000 MS/MS spectra contained in a single .mgf file (~2 h run time). After receiving input information, microbeMASST leverages the Fast Search Tool (https://fasst.gnps2.org/fastsearch/) API and the sample-specific associated metadata to generate taxonomic trees. Fast searches are based on indexing all the MS/MS spectra present in GNPS/MassIVE according to the mass and intensity of their precursor ions and then restricting the search to only a set of relevant spectra that have a greater chance of achieving a high spectral similarity (modified cosine score) to the MS/MS of interest. Searches are customizable and default settings are the following: precursor and fragment ion mass tolerances, 0.05; minimum cosine score threshold, 0.7; minimum number of matching fragment ions, 3; and analogue search, False. Users can modify these parameters on the basis of their data and research questions. Once matches are obtained, it is good practice to inspect the associated mirror plots for confirmation. To create the final taxonomic tree, the JSON file of the complete microbeMASST taxonomic tree is filtered according to the results and converted into a D3 JavaScript object that can be visualized as an HTML file.

### Applications

We envision microbeMASST to have several applications. First, we showcase how researchers can investigate single MS/MS spectra using the web interface and obtain matching results if their known or unknown MS/MS spectrum was previously observed in any of the microbial monocultures present in the microbeMASST database. Nine small molecules of interest were investigated using MS/MS spectra already deposited in the GNPS reference library (see 'Data availability' and 'Code availability'). Second, we show how microbeMASST can be leveraged to mine for known or unknown microbial metabolites in MS studies. To test this hypothesis, we reprocessed an LC–MS/MS dataset acquired from 29 different organs and biofluids of GF and SPF mice[30]. A comprehensive molecular network was generated (https://gnps.ucsd.edu/ProteoSAFe/status.jsp?task=893fd89b52dc4c07a292485404f97723). From the obtained job, the qiime2 artefact (qiime2_table.qza), the .mgf file (METABOLOMICS-SNETS-V2-893fd89b-

download_clustered_spectra-main.mgf) containing all the captured MS/MS spectra, and the annotation table (METABOLOMICS-SNETS-V 2-893fd89b-view_all_annotations_DB-main.tsv) were extracted. The .qza file was first converted into a .biom file and then a .tsv file using QIIME2 (ref. 60) to extract the feature table. This was then imported to R where only spectra present in tissues and biofluids of SPF animals were retained ($n = 10,047$). To add an extra layer of filtering, all MS/MS spectra that had an edge (cosine similarity >0.7) and a delta parent ion mass of ±0.02 Da with MS/MS spectra present in GF animals were removed. Spectral pairs information was contained in a networkedges_selfloop file. All the MS/MS spectra were then run in batch using a custom Python script of microbeMASST (processing time: ~2 h, Apple M2 Max, 64 GB RAM) to obtain microbial matches. Matching and filtered MS/MS spectra ($n = 2,425$) were aggregated into a single .mgf file that can be downloaded from GNPS (https://gnps.ucsd.edu/ProteoSAFe/status.jsp?task=aecd30b9febd43bd8f57b88598a05553). The compound class of each MS/MS spectrum with parent ion mass <850 Da was predicted using SIRIUS[38] and CANOPUS[39]. The 2,425 MS/MS spectra were then searched against the MSV000080918 dataset containing mice treated or not with antibiotics[40]. Matching and filtered MS/MS spectra ($n = 512$) were aggregated into a single .mgf file that can be downloaded from GNPS (https://gnps.ucsd.edu/ProteoSAFe/status.jsp?task=c33855fc32c948049331e9730189d5c1). A list of the spectra with putative chemical class classification is available in Supplementary Table 1. Venn diagrams and UpSet plots were generated in R using VennDiagram 1.7.3, UpSetR 1.4.0 and ComplexUpset 1.3.3. Finally, the 512 MS/MS spectra were searched in batch against the GNPS public repository to observe whether they were detected in human datasets (Supplementary Table 3). ReDU metadata information associated with the human datasets was then used to observe the distribution of the MS/MS spectra across different diseases and body parts.

### Technical limitations

Analysis of the results should be considered with the following limitations in mind. Molecule detection in microbeMASST is dependent on the availability of specific substrates in the reference monocultures. If the culture lacks the necessary substrates (or any other culture condition) to produce a certain molecule, this molecule will not be detected. Nevertheless, if related substrates are present, then a different but related molecule may be produced instead, which can be detected using the analogue search function. To address this problem, it is crucial for the community to continue to gather data from as many diverse experimental conditions as possible to capture the full range of metabolites produced by different microorganisms. This will help in building the most comprehensive reference database that encompasses diverse microbial metabolic profiles. In addition, isomers and stereoisomers, which are molecules with the same molecular formula but different structural arrangements, often exhibit similar MS/MS spectra. This means that their fragmentation patterns may not provide enough information to distinguish them. Finally, differences in extraction conditions and instrument settings can lead to variations in the obtained MS/MS spectra. For example, the intensity of precursor ions used for fragmentation can impact the resulting spectra. If the precursor ion intensity is low, the fragmented spectrum may lack ions that are present in spectra obtained from high-intensity precursor ions. This may result in 'data leakage' as the MS/MS spectrum may be missing ions, leading to the two molecules not being recognized as the same molecule. To partially overcome this, more permissive settings can be created. The majority of the data deposited in public repositories, GNPS included, and used in microbeMASST were acquired using positive electrospray ionization mode, which limits the observation of molecules that cannot be ionized in positive mode. This means that certain metabolites may be underrepresented or not detected at all. The continuous curation of the microbeMASST reference database involves adding more

diverse data in terms of ionization modes to improve the coverage of metabolites. The taxonomic tree was generated using associated NCBI IDs provided by the community. Specimen assignment before metabolomic analysis cannot be checked by microbeMASST. The majority of the deposited data do not have associated genetic information and even if available, it was not used to build the taxonomic tree. Thus, specimen misidentification cannot be excluded. By addressing these challenges and continuously curating the reference database with more comprehensive and diverse data, microbeMASST coverage can be expanded to provide valuable insights into the role of microbiota and to facilitate our understanding of microbial metabolism in diverse ecosystems.

### Statistics and reproducibility

No statistical method was used to predetermine sample size. No data were excluded from the analyses. The experiments were not randomized. The Investigators were not blinded to allocation during experiments and outcome assessment.

### Reporting summary

Further information on research design is available in the Nature Portfolio Reporting Summary linked to this article.

## Data availability

Data used to generate the reference database of microbeMASST are publicly available at GNPS/MassIVE (https://massive.ucsd.edu/). A list of all the accession numbers (MassIVE IDs) of the studies used to generate the reference database of this tool is available in Supplementary Table 2. Interactive examples of the MS/MS queries illustrated in Fig. 1d and Supplementary Fig. 3 can be visualized at the microbeMASST website (https://masst.gnps2.org/microbemasst/). A video tutorial on how to use microbeMASST is available on YouTube. Known molecules already present in the GNPS library (https://library.gnps2.org/) were used to facilitate interpretation and confirm that specific bacterial and fungal molecules of interest were exclusively observed in the respective monocultures.

Lovastatin - CCMSLIB00005435737
Salinosporamide A - CCMSLIB00010013003
Commendamide - CCMSLIB00004679239
Mevastatin - CCMSLIB00005435644
Arylomycin A4 - CCMSLIB00000075066
Yersiniabactin - CCMSLIB00005435750
Promicroferrioxamine - CCMSLIB00005716848
Glutamate-cholic acid (Glu-CA) - CCMSLIB00006582258
Glutamate-deoxycholic acid (Glu-DCA) - CCMSLIB00006582092

Data used to extract MS/MS spectra exclusively present in colonized (SPF) mice are publicly available in GNPS/MassIVE under the accession number MSV000079949. Data used to validate and assess antibiotics effect on microbial MS/MS spectra of interest are available under the accession number MSV000080918. A list of datasets with data acquired from human biosamples that presented matches to the putative microbial MS/MS spectra of interest is available in Supplementary Table 3.

## Code availability

The microbeMASST code to query spectra, create interactive trees and analyse results is available under an open-source license on GitHub (https://github.com/robinschmid/microbe_masst). This repository also contains code to run batch searches of thousands of MS/MS spectra by providing either a .tsv file containing a list of USIs or a .mgf file generated for example through the MZmine data processing pipeline. Code used to generate the microbeMASST web interface can be accessed on GitHub (https://github.com/mwang87/GNPS_MASST). Code used to perform the analysis and generate the figures presented in the manuscript can be downloaded from GitHub (https://github.com/simonezuffa/Manuscript_microbeMASST).

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

## Acknowledgements

This work was carried out through the collaborative microbial metabolite centre, which is supported by the National Institutes of Health (NIH) grant U24DK133658, the Alzheimer's gut project U19AG063744 and BBSRC-NSF award 2152526. We thank T. Adkins and L. McCormick from the USDA ARS Culture Collection for assistance in selecting and providing microbial strains used in this research. This project was supported in part by the US Department of Agriculture, Agricultural Research Service. A.M.C.-R. and H.M. were supported by NIH grant 1DP2GM137413. Research reported in this publication was supported in part by the National Center for Complementary and Integrative Health of the NIH under award number F32AT011475 to N.E.A. K.B.K. was supported by National Research Foundation of Korea (NRF) grants funded by the Korean Government (MSIT) (NRF-2020R1C1C1004046, 2022R1A5A2021216 and 2022M3H9A2096191). B.S. was supported by the Austrian Science Fund (FWF) P31915 and K.H. was financed by the FWF research group programme (grant FG3). D.P. was supported by the German Research Foundation (DFG) CMFI Cluster of Excellence (EXC 2124) and Collaborative Research Center CellMap (TRR 261). A.B., E.A.M., P.C.J., L.V.C.-L. and N.P.L. were supported by The São Paulo Research Foundation (FAPESP #2018/24865-4, #2019/03008-9, #2020/06430-0, #2022/12654-4, #2015/17177-6, #2020/02207-5, #2021/10603-0) and CNPq. C.L.-C. received financial support from StrainBiotech and the FEMSA Biotechnology Center from Tecnológico de Monterrey. L.R.-O. received a scholarship from the Mexican National Council of Science and Technology (CONACYT). W.H.G. was supported by NIH R01 GM107550. N.G. was supported by the NSF CAREER Award #2047235.

N.J.B., S.D. and J.S.S.D. were supported by ERC Horizon 2020 (grant agreement no. 694569) and by a Spinoza award (to J.S.S.D. from NWO). B.C.W. was supported by CMFI Cluster of Excellence (EXC 2124). M.B. was supported by DZIF (Grant no. TTU 09.722). A.V.P.d.L., S.L.L.R. and P.B.P. were supported by ERA-Net Cofund project BlueBio (grant agreement no. 311913), Research Council of Norway (300846). H.M.R., M.F.L. and G.L.B. were supported by Novo Nordisk Foundation (grant NNF19OC0056246; PRIMA—toward Personalized dietary Recommendations based on the Interaction between diet, Microbiome and Abiotic conditions in the gut). H.M.R. was supported by the Independent Research Fund Denmark (MOTILITY; grant no. 0171-00006B). E.C.O.N. was supported by a Nottingham Research Fellowship. A.I.P.-L. was supported by FPU (FPU19/00289). M.F.T. and R.d.C.P. were supported by R35GM12889. C.M.-S. was supported by Juan de la Cierva-Incorporación (IJC2018-036923-I) and Proyectos dirigidos por jóvenes investigadores de la Universidad de Málaga (B1-2021_21). D.R. was supported by Plan Nacional de I+D+i of the Ministerio de Ciencia e Innovación (PID2019-107724GB-I00) and Junta de Andalucía (P20_00479). X.L.G. was supported by a Nanyang Assistant Professorship. K.-S.J. was supported by NIH R01 GM137135. A.S. was supported by EMBO fellowship ALTF 996-2021. S.P. was supported by ETH Zurich Career Seed Fellowship. R.G. was supported by the Simons Foundation Postdoctoral Fellowship in Marine Microbial Ecology. H.C. was supported by NIH R01AI167860 and CIFAR. M.C.T. was supported by T32 DK007202 (NIDDK), the National Academies of Sciences, Engineering and Medicine through the Predoctoral Fellowship of the Ford Foundation, and the Howard Hughes Medical Institute (HHMI) Graduate Fellowships grant GT15123. E.E.-L. was supported by VI-Universidad de Costa Rica, Grants numbers C1604 and C0469. P.C. was supported by VI-Universidad de Costa Rica, Grants numbers C1604 and C0469; US National Science Foundation DEB-1638976. D.A.-V. and G.T.-C. were supported by VI-Universidad de Costa Rica, Grants numbers C1604 and C0469. K.F. was supported by CMFI Cluster of Excellence (EXC 2124). H.H.F.K. and G.F.d.S. were supported by Fundação de Amparo à Pesquisa do Estado do Amazonas (FAPEAM). D.B.S. was supported by Fundação de Apoio ao Desenvolvimento do Ensino, Ciência e Tecnologia do Estado de Mato Grosso do Sul - FUNDECT (process number: 71/032.390/2022, FUNDECT number: 311/2022). K.L.M. was supported by NIH/1R01GM132649. P.-M.A. was supported by a swissuniversities Open Research Data grant. M.C.M.K. and S.L.J. were supported by NIH R35GM142938. A.D.P. was supported by NIH U01 DK119702 and S10 OD021750. A.T.A. was supported by the Betty and Gordon Moore Foundation. M.L. and P.B. were supported by the Max Planck Society. Omnia Group Ltd. is duly acknowledged for microbial cultures. B.S.P. and N.B. were partially supported by NIH 1R01LM013115 and NSF ABI 1759980. R.K. was supported by NIH DP1AT010885. L.P.N. was supported by Omnia Group Ltd. We thank Shimadzu South Africa Ltd. for analytical support, and J. MacRae, head of The Metabolomics STP at the Francis Crick Institute, for guidance. J.-L.W. is supported by the Swiss National Science Foundation (SNSF) Bridge – Discovery 40B2-0_211759 for studies on fungal metabolomics.

## Author contributions

S.Z., R.S., A.B., M.W. and P.C.D. conceptualized the method. R.S., S.Z. and M.W. developed microbeMASST. P.C.D., S.Z., R.S., A.B., P.W.P.G., A.M.C.-R., Y.E.A., A.T.A., E.C.G., J. Zemlin, M.J.M., N.E.A., R.H.C., E.B., M.C.T., C.-Y.H., R.O., A.V.A., J. Zhao, H.C., M.C.M.K., S.L.J., F.T., L.P.N., N.E.M., I.A.D., E.A.M., L.V.C.-L., N.P.L., P.R.-T., P.C.J., B.R., A.D.P., M.F.T., R.d.C.P., G.T.-C., P.C., E.E.-L., D.A.-V., L.-M.Q.-G., J.-L.W., A.S., S.P., J.J., W.H.G., K.G., J.M.-C., P.-M.A., B.C.W., K.F., D.P., N.A., N.G., M.L., P.B., K.B.K., H.G., L.P.S.d.C., M.S.d.S., A.I.P.-L., C.M.-S., D.R., R.F., M.B., A.V.P.d.L., P.B.P., S.L.L.R., G.L.B., M.F.L., H.M.R., A.R., B.S., F.H., A.J.B., U.P., C.L.-C., L.R.-O., E.R., F.H., G.K., H.S., K.H., L.P., R.G., E.C.O.N., E.T.R., J.O., N.J.B., S.D., J.S.S.D., X.L.G., J.J.C., K.-S.J., D.B.S., F.M.R.S., G.F.d.S., H.H.F.K., C.G., J.A.C., H.M., K.B., K.L.M., S.E.O.-S., C.M.R., D.M. and R.K.

contributed data and curated metadata. S.Z. generated the taxonomic tree and performed analyses. R.S. developed the tree visualizer for enriched ontologies and output summaries. B.S.P. and N.B. developed the FASST algorithm. M.W. developed the Fast Search Tool API. S.Z., R.S. and P.C.D. tested microbeMASST. S.Z., R.S. and P.C.D. wrote the manuscript. All authors reviewed the manuscript.

## Competing interests

P.C.D. is an advisor to Cybele, consulted for MSD animal health in 2023 and is a co-founder and scientific advisor for Ometa Labs, Arome and Enveda, with previous approval from the University of California, San Diego. M.W. is a co-founder of Ometa labs. There are no known conflicts of interest in this work by the USDA, Agricultural Research Service, National Center for Agricultural Utilization Research, Mycotoxin Prevention and Applied Microbiology Research Unit. Mention of trade names or commercial products in this publication is solely for the purpose of providing specific information and does not imply recommendation or endorsement by the US Department of Agriculture. R.K. is a scientific advisory board member and consultant for BiomeSense, Inc., where he has equity and receives income. The terms of this arrangement have been reviewed and approved by the University of California, San Diego in accordance with its conflict-of-interest policies. The other authors declare no competing interests.

## Additional information

**Correspondence and requests for materials** should be addressed to Pieter C. Dorrestein.

Simone Zuffa [1,2,77], Robin Schmid [1,2,77], Anelize Bauermeister [1,2,3,77], Paulo Wender P. Gomes [1,2], Andres M. Caraballo-Rodriguez[1,2], Yasin El Abiead[1,2], Allegra T. Aron[4], Emily C. Gentry [5], Jasmine Zemlin[1,6], Michael J. Meehan[1], Nicole E. Avalon [7], Robert H. Cichewicz[8], Ekaterina Buzun[9], Marvic Carrillo Terrazas[9], Chia-Yun Hsu [9], Renee Oles[9], Adriana Vasquez Ayala[9], Jiaqi Zhao[9], Hiutung Chu [9,10], Mirte C. M. Kuijpers [11], Sara L. Jackrel[11], Fidele Tugizimana[12,13], Lerato Pertunia Nephali[12], Ian A. Dubery [12], Ntakadzeni Edwin Madala[14], Eduarda Antunes Moreira[15], Leticia Veras Costa-Lotufo[3], Norberto Peporine Lopes [15], Paula Rezende-Teixeira[3], Paula C. Jimenez[16], Bipin Rimal[17], Andrew D. Patterson [17], Matthew F. Traxler [18], Rita de Cassia Pessotti [18], Daniel Alvarado-Villalobos[19], Giselle Tamayo-Castillo [19,20], Priscila Chaverri[21,22,23], Efrain Escudero-Leyva [24], Luis-Manuel Quiros-Guerrero [25,26], Alexandre Jean Bory[25,26], Juliette Joubert[25,26], Adriano Rutz[25,26,27], Jean-Luc Wolfender [25,26], Pierre-Marie Allard [25,26,28], Andreas Sichert [27], Sammy Pontrelli [27], Benjamin S. Pullman[29], Nuno Bandeira[1,29], William H. Gerwick[1,7], Katia Gindro[30], Josep Massana-Codina [30], Berenike C. Wagner [31], Karl Forchhammer [31], Daniel Petras [32], Nicole Aiosa[33], Neha Garg[33,34], Manuel Liebeke [35,36], Patric Bourceau[35], Kyo Bin Kang [37], Henna Gadhavi [38,39], Luiz Pedro Sorio de Carvalho[38,40], Mariana Silva dos Santos [41], Alicia Isabel Pérez-Lorente[42], Carlos Molina-Santiago [42], Diego Romero[42], Raimo Franke [43], Mark Brönstrup [43,44], Arturo Vera Ponce de León[45], Phillip Byron Pope [45,46], Sabina Leanti La Rosa [45,46], Giorgia La Barbera [47], Henrik M. Roager [47], Martin Frederik Laursen [48], Fabian Hammerle[49], Bianka Siewert[49], Ursula Peintner[50], Cuauhtemoc Licona-Cassani[51], Lorena Rodriguez-Orduña [51], Evelyn Rampler[52], Felina Hildebrand [52,53], Gunda Koellensperger[52,54], Harald Schoeny [52], Katharina Hohenwallner [52,53], Lisa Panzenboeck [52,53], Rachel Gregor[55], Ellis Charles O'Neill [56], Eve Tallulah Roxborough[56], Jane Odoi[57], Nicole J. Bale[58], Su Ding [58], Jaap S. Sinninghe Damsté[58], Xue Li Guan [59], Jerry J. Cui [60], Kou-San Ju [60,61,62,63], Denise Brentan Silva[64], Fernanda Motta Ribeiro Silva[64], Gilvan Ferreira da Silva[65], Hector H. F. Koolen[66], Carlismari Grundmann[67], Jason A. Clement[68], Hosein Mohimani [69], Kirk Broders[70], Kerry L. McPhail [71], Sidnee E. Ober-Singleton[72], Christopher M. Rath[73], Daniel McDonald[74], Rob Knight[29,74,75], Mingxun Wang[76] & Pieter C. Dorrestein [1,2] ✉

[1]Skaggs School of Pharmacy and Pharmaceutical Sciences, University of California San Diego, San Diego, CA, USA. [2]Collaborative Mass Spectrometry Innovation Center, Skaggs School of Pharmacy and Pharmaceutical Sciences, University of California San Diego, San Diego, CA, USA. [3]Department of Pharmacology, Institute of Biomedical Sciences, University of São Paulo, São Paulo, Brazil. [4]Department of Chemistry and Biochemistry, University of Denver, Denver, CO, USA. [5]Department of Chemistry, Virginia Tech, Blacksburg, VA, USA. [6]Center for Microbiome Innovation, University of California San Diego, San Diego, CA, USA. [7]Scripps Institution of Oceanography, University of California San Diego, La Jolla, CA, USA. [8]Department of Chemistry and Biochemistry, College of Arts and Sciences, University of Oklahoma, Norman, OK, USA. [9]Department of Pathology, School of Medicine, University of California San Diego, San Diego, CA, USA. [10]Center for Mucosal Immunology, Allergy, and Vaccines (cMAV), Chiba University-University of California San Diego, San Diego, CA, USA. [11]Department of Ecology, Behavior and Evolution, School of Biological Sciences, University of California San Diego, San Diego, CA, USA. [12]Department of Biochemistry, Faculty of Science, University of Johannesburg, Johannesburg, South Africa. [13]International Research and Development, Omnia Nutriology, Omnia Group (Pty) Ltd, Johannesburg, South Africa. [14]Department of Biochemistry and Microbiology, Faculty of Sciences, Agriculture and Engineering, University of Venda, Thohoyandou, South Africa. [15]Department of BioMolecular Sciences, School of Pharmaceutical Sciences of Ribeirão Preto, University of São Paulo, Ribeirão Preto, São Paulo, Brazil. [16]Department of Marine Science, Institute of Marine Science, Federal University of São Paulo, Santos, Brazil. [17]Department of Veterinary and Biomedical Sciences, Pennsylvania State University, University Park, PA, USA. [18]Plant and Microbial Biology, College of Natural Resources, University of California Berkeley, Berkeley, CA, USA. [19]Metabolomics and Chemical Profiling, Centro de Investigaciones en Productos Naturales (CIPRONA), Universidad de Costa Rica, San José, Costa Rica. [20]Escuela de Química, Universidad de Costa Rica, San José, Costa Rica. [21]Microbial Biotechnology, Centro de Investigaciones en Productos Naturales (CIPRONA) and Escuela de Biología, Universidad de Costa Rica, San José, Costa Rica. [22]Escuela de Biología, Universidad de Costa Rica, San José, Costa Rica. [23]Department of Natural Sciences, Bowie State University, Bowie, MD, USA. [24]Microbial Biotechnology, Centro de Investigaciones en Productos Naturales (CIPRONA), Universidad de Costa Rica, San José, Costa Rica. [25]School of Pharmaceutical Sciences, University of Geneva, Geneva, Switzerland. [26]Institute of Pharmaceutical Sciences of Western Switzerland, University of Geneva, Geneva, Switzerland. [27]Institute of Molecular Systems Biology, ETH Zurich, Zurich, Switzerland. [28]Department of Biology, University of Fribourg, Fribourg, Switzerland. [29]Department of Computer Science and Engineering, University of California San Diego, San Diego, CA, USA. [30]Plant Protection, Mycology group, Agroscope, Nyon, Switzerland. [31]Department of Microbiology and Organismic Interactions, Interfaculty Institute of Microbiology and Infection Medicine, University of Tuebingen, Tuebingen, Germany. [32]Cluster of Excellence 'Controlling Microbes to Fight Infections' (CMFI), University of Tuebingen, Tuebingen, Germany. [33]School of Chemistry and Biochemistry, Georgia Institute of Technology, Atlanta, GA, USA. [34]Center for Microbial Dynamics and Infection, Georgia Institute of Technology, Atlanta, GA, USA. [35]Department of Symbiosis, Metabolic Interactions, Max Planck Institute for Marine Microbiology, Bremen, Germany. [36]Department for Metabolomics, Kiel University, Kiel, Germany. [37]Research Institute of Pharmaceutical Sciences, College of Pharmacy, Sookmyung Women's University, Seoul, Korea. [38]Mycobacterial Metabolism and Antibiotic Research Laboratory, The Francis Crick Institute, London, UK. [39]King's College London, London, UK. [40]Chemistry Department, The Herbert Wertheim UF Scripps Institute for Biomedical Innovation and Technology, Jupiter, FL, USA. [41]Metabolomics Science Technology Platform, The Francis Crick Institute, London, UK. [42]Department of Microbiology, Instituto de Hortofruticultura Subtropical y Mediterránea 'La Mayora', Universidad de Málaga-Consejo Superior de Investigaciones Científicas (IHSM-UMA-CSIC), Bulevar Louis Pasteur (Campus Universitario de Teatinos), Malaga, Spain. [43]Department of Chemical Biology, Helmholtz Centre for Infection Research, Braunschweig, Germany. [44]German Center for Infection Research (DZIF), Site Hannover-Braunschweig, Braunschweig, Germany. [45]Faculty of Chemistry, Biotechnology and Food Science, Norwegian University of Life Sciences, Ås, Norway. [46]Faculty of Biosciences, Norwegian University of Life Sciences, Ås, Norway. [47]Department of Nutrition, Exercise and Sports, University of Copenhagen, Frederiksberg, Denmark. [48]National Food Institute, Technical University of Denmark, Lyngby, Denmark. [49]Department of Pharmacognosy, Institute of Pharmacy, University of Innsbruck, Innsbruck, Austria. [50]Department of Microbiology, University of Innsbruck, Innsbruck, Austria. [51]Escuela de Ingeniería y Ciencias, Centro de Biotecnología FEMSA, Tecnologico de Monterrey, Monterrey, Mexico. [52]Department of Analytical Chemistry, Faculty of Chemistry, University of Vienna, Vienna, Austria. [53]Vienna Doctoral School in Chemistry (DoSChem), Faculty of Chemistry, University of Vienna, Vienna, Austria. [54]Vienna Metabolomics Center (VIME), University of Vienna, Vienna, Austria. [55]Department of Civil and Environmental Engineering, School of Engineering, Massachusetts Institute of Technology, Cambridge, MA, USA. [56]School of Chemistry, University of Nottingham, Nottingham, UK. [57]Faculty of Engineering, University of Nottingham, Nottingham, UK. [58]Department of Marine Microbiology and Biogeochemistry, Netherlands Institute for Sea Research (NIOZ), t Horntje (Texel), the Netherlands. [59]Lee Kong Chian School of Medicine, Nanyang Technological University, Singapore, Singapore. [60]Department of Microbiology, College of Arts and Sciences, The Ohio State University, Columbus, OH, USA. [61]Division of Medicinal Chemistry and Pharmacognosy, College of Pharmacy, The Ohio State University, Columbus, OH, USA. [62]Center for Applied Plant Sciences, The Ohio State University, Columbus, OH, USA. [63]Infectious Diseases Institute, The Ohio State University, Columbus, OH, USA. [64]Faculty of Pharmaceutical Sciences, Food and Nutrition, Federal University of Mato Grosso do Sul, Campo Grande, Mato Grosso do Sul, Brazil. [65]Embrapa Amazônia Ocidental, Manaus, Brazil. [66]Escola Superior de Ciências da Saúde, Universidade do Estado do Amazonas, Manaus, Brazil. [67]Department of Pharmaceutical Sciences, School of Pharmaceutical Sciences of Ribeirão Preto, University of São Paulo, Ribeirão Preto, Brazil. [68]Baruch S. Blumberg Institute, Doylestown, PA, USA. [69]Computational Biology Department, School of Computer Science, Carnegie Mellon University, Pittsburgh, PA, USA. [70]USDA, Agricultural Research Service, National Center for Agricultural Utilization Research, Mycotoxin Prevention and Applied Microbiology Research Unit, Peoria, IL, USA. [71]Department of Pharmaceutical Sciences, College of Pharmacy, Oregon State University, Corvallis, OR, USA. [72]Department of Physics, Study of Heavy-Element-Biomaterials, University of Oregon, Eugene, OR, USA. [73]Emeryville, CA, USA. [74]Department of Pediatrics, University of California San Diego, San Diego, CA, USA. [75]Department of Bioengineering, University of California San Diego, San Diego, CA, USA. [76]Department of Computer Science and Engineering, University of California Riverside, Riverside, CA, USA. [77]These authors contributed equally: Simone Zuffa, Robin Schmid, Anelize Bauermeister. ✉e-mail: pdorrestein@health.ucsd.edu

# Reporting Summary

## Statistics

For all statistical analyses, confirm that the following items are present in the figure legend, table legend, main text, or Methods section.

| n/a | Confirmed | |
|---|---|---|
| ☒ | ☐ | The exact sample size (*n*) for each experimental group/condition, given as a discrete number and unit of measurement |
| ☒ | ☐ | A statement on whether measurements were taken from distinct samples or whether the same sample was measured repeatedly |
| ☒ | ☐ | The statistical test(s) used AND whether they are one- or two-sided<br>*Only common tests should be described solely by name; describe more complex techniques in the Methods section.* |
| ☒ | ☐ | A description of all covariates tested |
| ☒ | ☐ | A description of any assumptions or corrections, such as tests of normality and adjustment for multiple comparisons |
| ☒ | ☐ | A full description of the statistical parameters including central tendency (e.g. means) or other basic estimates (e.g. regression coefficient) AND variation (e.g. standard deviation) or associated estimates of uncertainty (e.g. confidence intervals) |
| ☒ | ☐ | For null hypothesis testing, the test statistic (e.g. *F*, *t*, *r*) with confidence intervals, effect sizes, degrees of freedom and *P* value noted<br>*Give P values as exact values whenever suitable.* |
| ☒ | ☐ | For Bayesian analysis, information on the choice of priors and Markov chain Monte Carlo settings |
| ☒ | ☐ | For hierarchical and complex designs, identification of the appropriate level for tests and full reporting of outcomes |
| ☒ | ☐ | Estimates of effect sizes (e.g. Cohen's *d*, Pearson's *r*), indicating how they were calculated |

*Our web collection on statistics for biologists contains articles on many of the points above.*

## Software and code

Policy information about availability of computer code

| Data collection | Data extracted from the open access GNPS/MassIVE database using ReDU (update Dec 2021) and my manually inspecting entries in the MassIVE website |
|---|---|
| Data analysis | MicrobeMASST code is available at https://github.com/robinschmid/microbe_masst. Code of microbeMASST web application is available at https://github.com/mwang87/GNPS_MASST/blob/master/dash_microbemasst.py. Code related to microbeMASST manuscript analysis is available at https://github.com/simonezuffa/Manuscript_microbeMASST. Molecular networking was performed through GNPS (https://gnps.ucsd.edu/ProteoSAFe/static/gnps-splash.jsp) and metadata was gathered using ReDU (https://redu.ucsd.edu/). the Fast Search Tool is available at https://fasst.gnps2.org/fastsearch/. Used NCBI tools can be found at https://www.ncbi.nlm.nih.gov/Taxonomy/TaxIdentifier/tax_identifier.cgi and https://www.ncbi.nlm.nih.gov/Taxonomy/Browser/wwwtax.cgi. MicrobeMASST code and data analysis code are based on R 4.2.2 (R Foundation for Statistical Computing) and Python 3.10 (Python Software Foundation). |

For manuscripts utilizing custom algorithms or software that are central to the research but not yet described in published literature, software must be made available to editors and reviewers. We strongly encourage code deposition in a community repository (e.g. GitHub). See the Nature Portfolio guidelines for submitting code & software for further information.

## Data

Policy information about availability of data

All manuscripts must include a data availability statement. This statement should provide the following information, where applicable:

- Accession codes, unique identifiers, or web links for publicly available datasets
- A description of any restrictions on data availability
- For clinical datasets or third party data, please ensure that the statement adheres to our policy

> Data used to generate the reference database of microbeMASST are publicly available at GNPS/MassIVE (https://massive.ucsd.edu/). A list with the 537 accession codes (MassIVE IDs) of all the studies used to generate the reference database of this tool is available in Supplementary Table 2.

## Research involving human participants, their data, or biological material

Policy information about studies with human participants or human data. See also policy information about sex, gender (identity/presentation), and sexual orientation and race, ethnicity and racism.

| | |
|---|---|
| Reporting on sex and gender | No sex or gender are reported in the manuscript. A list of public datasets with data acquired from human biosamples that presented matches to the putative microbial MS/MS spectra of interest is available in Supplementary Table 3. |
| Reporting on race, ethnicity, or other socially relevant groupings | No race, ethnicity, or other socially relevant groupings are reported in the manuscript. A list of public datasets with data acquired from human biosamples that presented matches to the putative microbial MS/MS spectra of interest is available in Supplementary Table 3. |
| Population characteristics | No population characteristics is reported in the manuscript. A list of public datasets with data acquired from human biosamples that presented matches to the putative microbial MS/MS spectra of interest is available in Supplementary Table 3. |
| Recruitment | No recruitment is reported in the manuscript. A list of public datasets with data acquired from human biosamples that presented matches to the putative microbial MS/MS spectra of interest is available in Supplementary Table 3. |
| Ethics oversight | No ethics oversight is reported in the manuscript. A list of public datasets with data acquired from human biosamples that presented matches to the putative microbial MS/MS spectra of interest is available in Supplementary Table 3. |

Note that full information on the approval of the study protocol must also be provided in the manuscript.

# Field-specific reporting

Please select the one below that is the best fit for your research. If you are not sure, read the appropriate sections before making your selection.

☒ Life sciences    ☐ Behavioural & social sciences    ☐ Ecological, evolutionary & environmental sciences

For a reference copy of the document with all sections, see nature.com/documents/nr-reporting-summary-flat.pdf

# Life sciences study design

All studies must disclose on these points even when the disclosure is negative.

| | |
|---|---|
| Sample size | No sample size calculation was performed. All available data acquired from microbial monocultures deposited in GNPS/MassIVE were used to generate the microbeMASST reference database |
| Data exclusions | No data was excluded from the analysis |
| Replication | No biological replication required. Downstream data analysis can be replicated using publicly available code |
| Randomization | No randomization required and covariates were not controlled as not relevant to the study. Generate reference database encompass all relevant data deposited in GNPS/MassIVE and search tool is used to find matches between MS/MS spectra |
| Blinding | Blinding was not relevant to this study as we created a search tool. |

# Reporting for specific materials, systems and methods

We require information from authors about some types of materials, experimental systems and methods used in many studies. Here, indicate whether each material, system or method listed is relevant to your study. If you are not sure if a list item applies to your research, read the appropriate section before selecting a response.

## Materials & experimental systems

| n/a | Involved in the study |
|---|---|
| ☒ | Antibodies |
| ☒ | Eukaryotic cell lines |
| ☒ | Palaeontology and archaeology |
| ☒ | Animals and other organisms |
| ☒ | Clinical data |
| ☒ | Dual use research of concern |
| ☒ | Plants |

## Methods

| n/a | Involved in the study |
|---|---|
| ☒ | ChIP-seq |
| ☒ | Flow cytometry |
| ☒ | MRI-based neuroimaging |

## Plants

Seed stocks

*Report on the source of all seed stocks or other plant material used. If applicable, state the seed stock centre and catalogue number. If plant specimens were collected from the field, describe the collection location, date and sampling procedures.*

Novel plant genotypes

*Describe the methods by which all novel plant genotypes were produced. This includes those generated by transgenic approaches, gene editing, chemical/radiation-based mutagenesis and hybridization. For transgenic lines, describe the transformation method, the number of independent lines analyzed and the generation upon which experiments were performed. For gene-edited lines, describe the editor used, the endogenous sequence targeted for editing, the targeting guide RNA sequence (if applicable) and how the editor was applied.*

Authentication

*Describe any authentication procedures for each seed stock used or novel genotype generated. Describe any experiments used to assess the effect of a mutation and, where applicable, how potential secondary effects (e.g. second site T-DNA insertions, mosiacism, off-target gene editing) were examined.*

