## [Peer Review File · Nature Microbiology]

Peer Review Information

Journal: Nature Microbiology

Manuscript Title: MicrobeMASST: A Taxonomically-informed Mass Spectrometry Search Tool for Microbial Metabolomics Data

Corresponding author name(s): Professor Pieter Dorrestein

Reviewer Comments & Decisions:

Decision Letter, initial version:

Message: 11th September 2023

Dear Pieter,

Thank you for your patience while your manuscript "A Taxonomically-informed Mass Spectrometry Search Tool for Microbial Metabolomics Data" was under peer-review at Nature Microbiology. It has now been seen by 3 referees, whose expertise and comments you will find at the end of this email. Although they find your work of some potential interest, they have raised a number of concerns that will need to be addressed before we can consider publication of the work in Nature Microbiology.

In particular, referee #2 asks about the use of this tool for large existing datasets. Can people use this to annotate 100s-1000s spectra from existing datasets? Please move the section on this aspect of the tool from the SI to the main text, ensure that the code is accessible to referees and that this section is expanded and discussed since this will be the most useful aspect of the tool for the field. Additionally, the other referees have several technical questions that will need to be addressed. Finally, referee #3 notes in the confidential comments to the editor (which I will paraphrase) that the tool could not be used to identify metabolites that they have identified manually. Can you provide examples where MicrobeMASST can identify metabolites that have previously been elusive to existing tools? Or provide some explanation as to why the tool may not be able to identify some microbial metabolites.

Should further experimental data allow you to address these criticisms, we would be happy to look at a revised manuscript.

We strongly support public availability of data. Please place the data used in your paper into a public data repository, if one exists, or alternatively, present the data as Source Data or Supplementary Information. If data can only be shared on request, please explain why in your Data Availability Statement, and also in the correspondence with your editor. For some data types, deposition in a public repository is mandatory - more information on our data deposition policies and available repositories can be found at

2<https://www.nature.com/nature-research/editorial-policies/reporting-standards#availability-of-data>.

Please include a data availability statement as a separate section after Methods but before references, under the heading "Data Availability". This section should inform readers about the availability of the data used to support the conclusions of your study. This information includes accession codes to public repositories (data banks for protein, DNA or RNA sequences, microarray, proteomics data etc...), references to source data published alongside the paper, unique identifiers such as URLs to data repository entries, or data set DOIs, and any other statement about data availability. At a minimum, you should include the following statement: "The data that support the findings of this study are available from the corresponding author upon request", mentioning any restrictions on availability. If DOIs are provided, we also strongly encourage including these in the Reference list (authors, title, publisher (repository name), identifier, year). For more guidance on how to write this section please see: <http://www.nature.com/authors/policies/data/data-availability-statements-data-citations.pdf>

* If you have not done so already we suggest that you begin to revise your manuscript so that it conforms to our Brief Communication format instructions at <http://www.nature.com/nmicrobiol/info/final-submission>. Refer also to any guidelines provided in this letter.

When submitting the revised version of your manuscript, please pay close attention to our [href="https://www.nature.com/nature-portfolio/editorial-policies/image-integrity">Digital Image Integrity Guidelines. and to the following points below:](https://www.nature.com/nature-portfolio/editorial-policies/image-integrity)

Finally, please ensure that you retain unprocessed data and metadata files after publication, ideally archiving data in perpetuity, as these may be requested during the peer

2review and production process or after publication if any issues arise.

Note: This url links to your confidential homepage and associated information about manuscripts you may have submitted or be reviewing for us. If you wish to forward this e-mail to co-authors, please delete this link to your homepage first.

Nature Microbiology is committed to improving transparency in authorship. As part of our efforts in this direction, we are now requesting that all authors identified as 'corresponding author' on published papers create and link their Open Researcher and Contributor Identifier (ORCID) with their account on the Manuscript Tracking System (MTS), prior to acceptance. This applies to primary research papers only. ORCID helps the scientific community achieve unambiguous attribution of all scholarly contributions. You can create and link your ORCID from the home page of the MTS by clicking on 'Modify my Springer Nature account'. For more information please visit www.springernature.com/orcid.

If you wish to submit a suitably revised manuscript we would hope to receive it within 6 months. If you cannot send it within this time, please let us know. We will be happy to consider your revision, even if a similar study has been accepted for publication at Nature Microbiology or published elsewhere (up to a maximum of 6 months).

Yours sincerely,

Reviewer Expertise:

Referee #1: metabolomics, microbiome
Referee #2: metabolomics, bioinformatics
Referee #3: metabolomics, bioinformatics

Reviewer Comments:

Reviewer #1 (Remarks to the Author):

Zuffa et al. described a tool, MicrobeMASST, which allows researchers to identify potential metabolites and the potential origins of these metabolites/MS-MS signals from a database containing metabolome information from cultures of bacteria, archaea, fungi and human cell lines. MicroMASST would add significant value to the microbiome research field, in particular, addressing the metabolic function of the microbes. While a few examples were described and the data interpretation was given, the limitation of the tool and the caution of data interpretation based on the information provided by the database should be highlighted in these examples rather than in Method, e.g. MS/MS match does not guarantee confirmation of a metabolite, signals that are not present in the microorganism

3or human cell line culture do not necessarily indicate they cannot be produced by them, caution should be paid when claiming the sole origins of the metabolites. Specifically, "Based on literature information, the list of annotated MS/MS spectra contained a small number of metabolites traditionally considered to be non-microbial in origin." Some examples from this small number of metabolites should be given here. It is unclear if this list is the new finding from the tool. If so, examples of metabolites given below (erotonin, γ -aminobutyric acid (GABA), and the glycocholic acid) are not supportive to the finding since these metabolite are known to be produced by both the mammalian host and bacteria. "... with microorganisms often being the primary producers of these metabolites in the gut." References should be cited to indicate microorganisms are the primary producers rather than the mammalian host. What does the 'gut' refer to here, the gut lumen or gut tissue?

Reviewer #2 (Remarks to the Author):

The author present taxonomically informed search tool for MS/MS spectra of metabolites (known and unknown metabolites), with the aim to facilitate annotation of metabolites of microbial origin. Such a tool is highly useful for the annotation of untargeted metabolomics data, where the challenge is to determine the origin of specific unannotated metabolites (e.g., in human studies, if metabolites are endogenously produced by the host, or by the [gut] microbes). The tool will be very useful to microbiologists and other researchers utilizing metabolomics in their research.

The paper is well written and presentation is good. The tool itself is integrated with GNPS, easy to use and fast due to use of Fast Search Tool.

Three comments to consider:

1. In order for the microbeMASST to be useful in metabolomics studies, it would be important that the tool could annotate datasets from metabolomics studies. This is indeed possible through batch processing, but this is only briefly mentioned in the supplement, referring to Python code that does not seem to be available.
2. Is it foreseen that microbeMASST could be integrated with the data processing softwares, e.g., MZmine? There is no mention how the tool could be used as part of the integrated data processing pipeline.
3. With regards to the murine study used as an example (Fig. 2), the study is properly referenced and there is sufficient detail in the supplement. However, in main manuscript, it would be useful to know where the spectra are coming from (specific tissues, biofluids).

Reviewer #3 (Remarks to the Author):

Lack of microbial metabolite databases with mass spectra, including fragmentation data is a major problem in the field. The ability to gather all this public MS data and allow for matching for both metabolite and species/taxa will greatly benefit the field. The authors have also done a great job constructing the manuscript and the limitations section is important and also well written. I just have some minor comments that are mostly to improve clarity, especially if the authors wish to have a greater reach to new users of

4GNPS/microbeMASST portal.

1) Some of the descriptions of numbers of LC-MS/MS files and MS/MS spectra are a little confusing and could be clarified. For example when it is mentioned "each MS/MS query is searched against the public MS/MS reference library of GNPS, this wasn't quite clear whether the GNPS library vs. the microbeMASST libraries contain different data or a subset of just microbial data?

2) There is also some assumption of prior knowledge of MASST and GNPS for the reader. It is written that microbeMASST works within GNPS, therefore how does using the GNPS system to search for an entry, differ from using the microbeMASST system, is it the link to the taxa? It also seems that the advance from MASST to microbeMASST is the speed and taxonomy link, is the datasource the same? Perhaps this could be clarified.

3) For the interface could the authors explain more what a USI refers to and how it was constructed, I believe this is a term introduced by the proteomics field 2 years ago but not commonly used in metabolomics? Is this specific to GNPS or is it used outside of this software? Also what do the inputs and outputs mean on the system, and are there recommended thresholds for matching based on the mass accuracy of the analysis performed by the contributors? without apriori knowledge. For example for charge it seems this is the number of charges vs a negative or positive charge. It seems like ESI mode might be important as well for generating a match, but it doesn't appear this is an input anywhere?

4) When there is more than one metabolome curated for a given microbe, I see that there are available samples and matching samples listed. If the MS/MS spectrum that one enters matches to 50% of that microbe (represented as the pie chart), is that a complete match to half of the available samples? Or are X number of peaks matched similarly in that half of the samples. This could be the explanation for the taxa matches tab but it isn't clear – could it also be explained what all the outputs mean on this tab as well?

5) It would be good to mention that the matches are to partly or fully unidentified metabolites.

6) In general the font sizes are too small on the figures when reading the paper at 100% zoom.

Author Rebuttal to Initial comments

Editor Comment:

Dear Pieter,

Thank you for your patience while your manuscript "A Taxonomically-informed Mass Spectrometry Search Tool for Microbial Metabolomics Data" was under peer-review at Nature Microbiology. It has now been seen by 3 referees, whose expertise and comments you will find at the end of this email. Although they find your work of some potential interest, they have raised a number of concerns that will need to be addressed before we can consider publication of the work in Nature Microbiology.

In particular,

referee #2 asks about the use of this tool for large existing datasets. Can people use this to annotate 100s-1000s spectra from existing datasets? Please move the section on this aspect of the tool from the SI to the main text, ensure that the code is accessible to referees and that this section is expanded and discussed since this will be the most useful aspect of the tool for the field.

Thanks for the feedback summary. Users can indeed search for as many MS/MS spectra as they wish. We have provided the code to do batch MS/MS searches (https://github.com/robinschmid/microbe_masst). The code was already accessible at the moment of submission, but it was not clearly explained in the text. This is now clearly stated in lines 179-183.

Additionally, the other referees have several **technical questions that will need to be addressed**.

We thank the referees for the great comments and suggestions. All technical questions have been addressed.

Finally, referee #3 notes in the confidential comments to the editor (which I will paraphrase) that the **tool could not be used to identify metabolites that they have identified manually. Can you provide examples where MicrobeMASST can identify metabolites that have previously been elusive to existing tools? Or provide some explanation as to why the tool may not be able to identify some microbial metabolites.**

If we understand correctly, the referee searched for an MS/MS spectrum that they know to be of microbial origin, but no match was found through microbeMASST. Without knowing more details it is difficult to completely understand the issue they faced but it could be due to one of the limitations we discussed in the Technical limitations section of the paper. 1) The metabolite is produced exclusively by a microorganism that is missing from the reference database of microbeMASST. 2) If the microorganism is present in the reference database, it might be possible that the data acquired from the different culture conditions does not present the metabolite for a series of reasons that have also been discussed in the Technical limitation section of the manuscript. For example, A) under the analyzed culture conditions the microorganism did not produce the metabolite of interest although it is capable of doing so. B) The extraction and LC-MS methods used in all available public metabolomics experiments were not tailored to the extraction of the metabolite of interest. 3) Finally, data of the referee and data in the microbeMASST database could have been acquired with different instruments, with different mass accuracy and intensities for example, and “data leakage” (see detailed explanation in the manuscript, line 690) might result in two spectra not being recognized as the same one. The referee could try to increase tolerance of parent and fragment ion masses, decrease the cosine similarity threshold, or decrease the minimum number of matched fragment peaks. More permissive settings could allow the referee to find a match, but false positives could also appear. The linked mirror plots aid in validating the findings.

If for identification of metabolite they meant annotation, microbeMASST is not meant for annotation but exclusively to link MS/MS spectra to microbial producers. We do provide annotation by matching the queries MS/MS spectrum to the public GNPS library and if the metabolite of interest is not present in it, or if the reference spectrum is collected under different fragmentation conditions (generating different fragment ions), no annotation will be provided.

A general remark is that both the GNPS library and the microbeMASST reference database of microbial extracts are open resources and we expect them to grow through further community contributions. If the reviewer is able to share their datasets and spectra of identified compounds, we are happy to facilitate their addition to microbeMASST and GNPS library and/or assess the issue they faced.

We hope that this answers the confidential question of referee #3. Please let us know if additional clarifications are required.

Reviewer #1 (Remarks to the Author):

Zuffa et al. described a tool, MicrobeMASST, which allows researchers to identify potential metabolites and the potential origins of these metabolites/MS-MS signals from a database containing metabolome information from cultures of bacteria, archaea, fungi and human cell lines. **MicroMASST would add significant value to the microbiome research field, in particular, addressing the metabolic function of the microbes.**

We thank the reviewer for the kind words, and we are looking forward to seeing how the metabolomics community will use this tool to improve our knowledge on microbial derived metabolites.

While a few examples were described and the data interpretation was given, the limitation of the tool and the caution of data interpretation based on the information provided by the database **should be highlighted in these examples rather than in Method**, e.g. MS/MS match does not guarantee confirmation of a metabolite, signals that are not present in the microorganism or human cell line culture do not necessarily indicate they cannot be produced by them, caution should be paid when claiming the sole origins of the metabolites.

We agree with the reviewer. We aimed to be as clear as possible in describing the technical limitations of microbeMASST, which are inherent to mass spectrometry. As pointed out by the reviewer, limitations are discussed in the Methods section (Technical limitations) and now are also presented in the main text (lines 211-217). As mentioned by the reviewer, MS/MS match indeed does not guarantee confirmation of a metabolite identification (level 1 annotation) but a level 2 annotation, which was also mentioned in the original manuscript in line 288 with reference to the Metabolomics Standards Initiative (.. Of these 512 spectra, 23% had a level 2 annotation, matching against the GNPS reference libraries ..). We have now added to lines 288-291 "... level 2 putative annotation according to the 2007 metabolomics standards initiative ... A level 2 annotation, within the user specified search criteria, might result in MS/MS matches between molecules belonging to related families as opposed to unique molecules."

In the main section we have now written (lines 211-217):

It is important to point out that microbeMASST allows to link both partly annotated, through MS/MS match to a reference library spectrum, and fully uncharacterized spectra to possible microbial producers but that technical limitations inherent in mass spectrometry or the experiment itself are present. For example, the absence of a matching spectrum in a specific taxon does not necessarily indicate that it is not capable of producing that molecule but rather that the methodology used to acquire the data did not allow its detection. These and other limitations are described in the Methods section. Despite these limitations, microbeMASST uniquely enables the discovery of links of uncharacterized MS/MS spectra to defined microorganisms, providing valuable information for future mechanistic studies.

Specifically, "Based on literature information, the list of annotated MS/MS spectra contained a small number of metabolites traditionally considered to be non-microbial in origin." **Some examples from this small number of metabolites should be given here. It is unclear if this list is the new finding from the tool.** If so, examples of metabolites given below (serotonin, γ -aminobutyric acid (GABA), and the glycocholic acid) are not supportive to the finding since these metabolites are known to be produced by

both the mammalian host and bacteria. "... with microorganisms often being the primary producers of these metabolites in the gut."

We apologize for the confusing statement. We hope this tool will enable users in the microbial and metabolomics community to find and describe new microbial molecules. The sentence "... Based on literature information, the list of annotated MS/MS spectra contained a small number of metabolites traditionally considered to be non-microbial in origin." was referring to the annotated di- and tri-hydroxy bile acids and glycine conjugated bile acids. Based on the reviewer comment we have now changed the sentence to "... This list of annotated MS/MS spectra include metabolites that are not yet widely considered to be of microbial origins, such as di- and tri-hydroxylated bile acids as well as glycine conjugated bile acids". Although probably not clear in the text (which has been now clarified), serotonin, GABA, and glycocholic acid are reported as examples of metabolites that were previously considered exclusively produced by the mammalian host but that today are known to be produced by bacteria too. Cholesterol has also been recently shown to be produced de novo by bacteria (<https://www.nature.com/articles/s41467-023-38638-8>). The manuscript already present references supporting the evidence that bacteria can produce serotonin ([https://www.cell.com/fulltext/S0092-8674\(15\)00248-2](https://www.cell.com/fulltext/S0092-8674(15)00248-2)), GABA (<https://www.nature.com/articles/s41564-018-0307-3>), and glycocholic acid (https://journals.asm.org/doi/full/10.1128/msystems.00805-21?rfr_dat=cr_pub++0pubmed&url_ver=Z39.88-2003&rfr_id=ori%3Arid%3Acrossref.org).

Glycocholic acid was only recently described to be produced by gut bacteria and the scientific community probably still consider it exclusively host-derived. Additionally, we observed several conjugated bile acids with amino acids (arginine, glycine, leucine/isoleucine, lysine, phenylalanine, serine, and tyrosine) which the research community might still not be familiar with. We have recently described them in complex biosamples (<https://www.nature.com/articles/s41586-020-2047-9>, <https://www.researchsquare.com/article/rs-820302/v1>) and only in the past year we showed that bacterial bile salt hydrolase (BSH) is responsible for the amino acid conjugation (<https://www.researchsquare.com/article/rs-2050120/v1>).

In the main section we have now written (lines 298-300):

This list of annotated MS/MS spectra include metabolites that are not yet widely considered to be of microbial origins, such as di- and tri-hydroxylated bile acids as well as glycine conjugated bile acids.

And (lines 302-303):

Notable examples of metabolites that have been established to be produced by both the mammalian host and bacteria include serotonin ...

References should be cited to indicate microorganisms are the primary producers rather than the mammalian host. What does the 'gut' refer to here, the gut lumen or gut tissue?

We apologize for the confusing statement as this was not clear and ambiguity remains. The sentence was referring to serotonin being primary produced by the gut tissue (95% of total body serotonin) under the regulation on the gut microbiome via short chain fatty acids and secondary bile acids (<https://www.nature.com/articles/s41398-022-01922-0>, <https://www.sciencedirect.com/science/article/pii/S0958166922001604>). As actually this is not strictly pertinent to the discussion of the microbeMASST tool, we have now removed the sentence.

Reviewer #2 (Remarks to the Author):

The author presents a taxonomically informed search tool for MS/MS spectra of metabolites (known and unknown metabolites), with the aim to facilitate annotation of metabolites of microbial origin. **Such a tool is highly useful for the annotation of untargeted metabolomics data**, where the challenge is to determine the origin of specific unannotated metabolites (e.g., in human studies, if metabolites are endogenously produced by the host, or by the [gut] microbes). **The tool will be very useful to microbiologists and other researchers utilizing metabolomics in their research. The paper is well written and the presentation is good.** The tool itself is integrated with GNPS, easy to use and fast due to use of Fast Search Tool.

We thank the reviewer for the kind words, and we are looking forward to seeing how the metabolomics community will use this tool to improve our knowledge on microbial derived metabolites.

Three comments to consider:

1. In order for the microbeMASST to be useful in metabolomics studies, it would be important that the tool could annotate datasets from metabolomics studies. This is indeed possible through batch processing, but this is **only briefly mentioned in the supplement, referring to Python code that does not seem to be available.**

11We thank the reviewer for pointing this out. The Python code for batch searches was already available at the moment of submission (https://github.com/robinschmid/microbe_masst) and was used to search >10,000 MS/MS spectra obtained from the GF vs SPF dataset. This was not clearly stated in the manuscript and now, lines 179-184 state that batch searches can be conducted on thousands of MS/MS spectra using custom Python code. Additionally, in the Methods section lines 560-563 now clearly state that the repository above mentioned (https://github.com/robinschmid/microbe_masst) also contains the script to run batch searches. Link to the repository has also been added in line 624 for extra clarity.

In the main text we have now written (lines 179-184):

Finally, users can leverage microbeMASST Python code to perform batch searches of thousands of MS/MS spectra by providing either a formatted MS/MS file (.mgf) or a list of Universal Spectrum Identifiers (USIs)²¹, which represent paths to spectra in public datasets. This is particularly useful for creating integrated data analysis pipelines using the standard outputs (.mgf) of already established data processing tools, such as MZmine.

As well as in the code availability section we have now written (lines 560-563):

This repository also contains code to run batch searches of thousands of MS/MS spectra by providing either a .tsv file containing a list of USIs or an .mgf file generated for example through the MZmine data processing pipeline.

2. Is it foreseen that microbeMASST could be integrated with the data processing softwares, e.g., MZmine? **There is no mention how the tool could be used as part of the integrated data processing pipeline.**

Thanks for this great suggestion. MicrobeMASST can be easily integrated in automated data processing pipelines as it can read as input the .mgf file generated by MZmine or the GNPS classical molecular networking without any need for modifications. Lines 179-183 now state that microbeMASST can be used as part of data processing pipelines using the output of tools such as MZmine.

In the main section we have now written (lines 179-183):

Finally, users can leverage microbeMASST Python code to perform batch searches of thousands of MS/MS spectra by providing either a formatted MS/MS file (.mgf) or a list of Universal Spectrum

Identifiers (USIs)²¹, which represent paths to spectra in public datasets. This is particularly useful for creating integrated data analysis pipelines using the standard outputs (.mgf) of already established data processing tools, such as MZmine.

3. With regards to the murine study used as an example (Fig. 2), the study is properly referenced and there is sufficient detail in the supplement. However, in the main manuscript, **it would be useful to know where the spectra are coming from** (specific tissues, biofluids).

We agree with the reviewer, as it is important to mention that the study we re-analyzed comprises MS data acquired from 29 different organs collected from germ-free and SPF mice, one of the most tissue-wise comprehensive metabolomics studies publicly available! Line 271-272 now states that the data was acquired from 29 different organs and biofluids, and some examples are provided (brain, heart, liver, blood, and stool). A complete list of organs that have been analyzed can be found in the referenced paper and in the metadata file associated with the study repository that has already been linked in the 2020 Quinn et al manuscript (MSV000079949).

In the main text we have now written (lines 271-272):

.. with data acquired from 29 different organs and biofluids, comprising tissues including brain, heart, liver, blood, and stool, ...

Reviewer #3 (Remarks to the Author):

Lack of microbial metabolite databases with mass spectra, including fragmentation data is a major problem in the field. The ability to gather all this public MS data and allow for matching for both metabolite and species/taxa **will greatly benefit the field**. The authors have also done a **great job constructing the manuscript and the limitations section is important and also well written**. I just have some **minor comments that are mostly to improve clarity**, especially if the authors wish to have a greater reach to new users of GNPS/microbeMASST portal.

We thank the reviewer for their time, comments on clarity and recognizing the value of this unique community-based resource.

1) **Some of the descriptions of numbers of LC-MS/MS files and MS/MS spectra are a little confusing and could be clarified**. For example when it is mentioned “each MS/MS query is searched against the public MS/MS reference library of GNPS, this wasn’t quite clear whether the GNPS library vs. the microbeMASST libraries contain different data or a subset of just microbial data?”

Thanks for pointing this out, as we want to be as clear as possible to also reach users who are not familiar with the GNPS environment. A queried MS/MS spectrum is searched against both the reference library of GNPS, which contains ~600K MS/MS reference spectra whose structures are known, and all the publicly available datasets deposited in GNPS/MassIVE through the years (currently ~2800 public projects). Matching against all public MS2 spectra from those data sets results in a list of samples that is then matched against the microbeMASST reference database. This allows us to select only samples acquired from microbial monocultures and to enrich them with detailed microbial taxonomy. For this purpose, a subset of microbial related studies has been curated, together with the community, to contain detailed metadata that ultimately allows the generation of the taxonomic tree and the utilization of data exclusively obtained from monocultures of microorganisms, as described in the manuscript. A list of the accession numbers of the datasets that have been used to generate the reference database of microbeMASST is available in the Methods section (Supplementary Table 2). Lines 157-160 now state

14*that the search of the MS/MS spectrum is performed against the whole GNPS/MassIVE repository and then only samples acquired from microbial monocultures are presented to the user. Details in data curation and harmonization of the microbial datasets are discussed in the **Methods** section. Consequently, we have updated the text in line 186 to state that when a spectrum search is performed only matching samples present in the reference database of microbeMASST are reported (.. can search single MS/MS spectra and obtain matching results from the reference database ..) and in line 200 to mention that the search is also performed against the public library of GNPS to provide annotation when available.*

In the main text we have now written (lines 157-160):

Users can search tandem MS (MS/MS) spectra obtained from their experiments against the GNPS/MassIVE repository and retrieve matching samples exclusively acquired from extracts of bacterial, fungal, or archaeal monocultures.

Please let us know if any additional clarification in the main text is needed.

2) There is also some assumption of prior knowledge of MASST and GNPS for the reader. It is written that microbeMASST works within GNPS, therefore **how does using the GNPS system to search for an entry differ from using the microbeMASST system**, is it the link to the taxa? It also seems that the advance from MASST to microbeMASST is the speed and taxonomy link, **is the data source the same?** Perhaps this could be clarified.

Yes, the use of microbeMASST requires a basic knowledge of GNPS and mass spectrometry concepts (MS/MS, cosine similarity, mass accuracy) to understand where the data is stored, but not to run it, and what the search parameters mean. GNPS is a fully fledged web-platform with many different data processing workflows, community resources, and links to many other third party tools. We do recognize that someone not familiar with these concepts may have to learn about them. Nevertheless, we are very proud of the extensive documentation that is publicly available for GNPS (<https://ccms-ucsd.github.io/GNPSDocumentation/>) and the many workshops available for free on YouTube - even in different languages (CMFI Mass Spec Seminar #5 - Intro to Molecular Networking, GNPS FBMN in Portuguese Part 1). Additionally several publications are available (<https://www.nature.com/articles/nbt.3597>, <https://www.nature.com/articles/s41592-020-0933-6>) including a Nature Protocols paper (<https://www.nature.com/articles/s41596-020-0317-5>) that the users can read to get familiar with the concepts at the foundations of the GNPS environment.

The data source for microbeMASST is the same as MASST - all public metabolomics studies deposited in GNPS/MassIVE. Search results are refined using microbeMASST database of known microbial monocultures extracts and mapped to a taxonomic tree. Texted has been updated in response to the previous question to be clearer (lines 157-160). Improvements of microbeMASST over MASST are described in lines 170-183. We introduced MASST (<https://www.nature.com/articles/s41587-019-0375-9>) in 2020, which allows users to search for a single spectrum against the GNPS/MassIVE repository of public datasets. This presents several limitations as a single search usually takes up to 20 minutes to be performed or 24-48 hours for modification tolerant searches. Additionally, MASST results are reported in an unstructured way (i.e. a list of file paths that have a match to the queried spectrum), and it is up to the user to search manually for each of them and understand what kind of samples they are. Additionally, MASST is not restricted to a subset of data, so the results are mixed with all kinds of samples and studies. MicrobeMASST solves these problems as searches are performed within seconds. We use the Fast Search Tool API to search all the MS/MS spectra available in GNPS/MassIVE. We then take the list of matching samples, and we match it against a list of curated datasets that contain only data acquired from monocultures of microorganisms. Each of these files has a paired curated metadata that allows to generate a taxonomic tree for easy interpretation and visualization. Additionally, the implemented fast search has unlocked the possibility for users to perform batch MS/MS submissions via custom Python code (now mentioned in lines 179-183) to search thousands of MS/MS spectra in just a few hours, something that was impossible before. It would have taken one year to search for 1,000 spectra while now we searched 10,000 in just 2 hours.

3) For the interface, could the authors **explain more what a USI refers to and how it was constructed?** I believe this is a term introduced by the proteomics field 2 years ago but not commonly used in metabolomics? **Is this specific to GNPS or is it used outside of this software?**

Universal Spectrum Identifiers (USIs) represent a standardized encoding path to any mass spectrum contained in a given metabolomics repository (<https://www.nature.com/articles/s41592-021-01184-6>, <https://www.biorxiv.org/content/10.1101/2020.05.09.086066v2.abstract>), information now added to line 181. USIs are extensively used in the GNPS environment as they represent a quick and convenient way to visualize and inspect mass spec data or to share links to specific datasets. For example, they can be used in the GNPS dashboard (<https://dashboard.gnps2.org/>) to explore your MS sample (<https://www.nature.com/articles/s41592-021-01339-5>). Importantly, USIs are not limited to GNPS/MassIVE as they are supported by the other two major metabolomics data repositories, Metabolights and Metabolomics Workbench. Also, desktop tools like MZmine make use of USI by accessing spectral data through a web API. USIs examples for different repositories are the following:

16GNPS/MassIVE - mzspec:MSV000084494:GNPS00002_A3_p; Metabolights - mzspec:MTBLS2700:CTRL_Gut_mzXML/M_sexta_21d_CTRL_1.mzXML; and Metabolomics Workbench - mzspec:ST002807:Conversion_MZXML20220202_B10_1j_M1_25_1_610.mzXML. USIs can also be easily generated using the web interface in the GNPS Explorer (<https://explorer.gnps2.org/>). Given the restricted number of words for the manuscript format we believe that there is no need to describe USIs in the manuscript as readers can learn more about them from the referenced paper (<https://www.nature.com/articles/s41592-021-01184-6>, <https://www.biorxiv.org/content/10.1101/2020.05.09.086066v2.abstract>).

In the main section we have now written (line 181):

.. or a list of Universal Spectrum Identifiers (USIs), which represent paths to spectra in public datasets.

4) Also what do the inputs and outputs mean on the system, and are there **recommended thresholds for matching based on the mass accuracy of the analysis performed by the contributors?** without a priori knowledge. For example for charge it seems this is the number of charges vs a negative or positive charge. It seems like ESI mode might be important as well for generating a match, but it doesn't appear this is an input anywhere?

The default thresholds are the ones that are already preset by default in the microbeMASST web interface upon opening (precursor mass tolerance 0.05 Da, fragment ion tolerance 0.05 Da, cosine score 0.7 and minimum matched peaks 3). This was not explicitly discussed and is now stated in line 632. These parameters can be modified by the users as they wish, based on their data and their research question. For example, if a user has acquired their data at high resolution (orbitrap) and they may want to conduct a stricter search, they can lower the tolerances of the parent and fragment ions mass errors to 0.01. Also, if they are searching for bigger molecules, the number of minimum matching peaks can also be raised to 5 or 7 or even more. Although we suggest keeping the cosine score similarity to 0.7, this can also be lowered if the users consider it appropriate. Users should then inspect the generated mirror plots to make sure that the matching spectra are indeed similar (now stated in line 635).

The GNPS/MassIVE repository contains LC-MS/MS data acquired on different instruments with different level of mass accuracy (i.e. QTOF vs Orbitrap) and 90% of the data deposited in GNPS/MassIVE has been acquired in ESI positive mode. Similar numbers are also true for the metabolomics data repositories, Metabolights and Metabolomics Workbench. This has been discussed in the Technical

limitation section (lines 692-695). As the mass difference between a protonated and deprotonated MS/MS is 2 Da, there should not be a match between the two spectra. When library matches are obtained (see figures below) the charge and adducts are displayed so the user can confirm if these match their ionization.

Library matches		Dataset matches		Taxa matches		Parameters	
Copy	Excel	CSV	Show	4	entries	Search: <input type="text"/>	
Delta Mass	GNPSLibraryAccession	USI	Charge	Cosine	Matching Peaks	CompoundName	Adduct
0	CCMSLIB00005435737	mzspec:GNPS.GNPS-LIBRARY.accession:CCMSLIB00005435737	1	1	12	Lovastatin M+H; Mevinolin annotated in standard	M+H
0	CCMSLIB00006116679	mzspec:GNPS.GNPS-LIBRARY.accession:CCMSLIB00006116679	1	1	12	Lovastatin - 40.0 eV	M+H
0	CCMSLIB00006116681	mzspec:GNPS.GNPS-LIBRARY.accession:CCMSLIB00006116681	1	0.96	11	Lovastatin - 40.0 eV	M+H
0	CCMSLIB00006116683	mzspec:GNPS.GNPS-LIBRARY.accession:CCMSLIB00006116683	1	0.96	11	Lovastatin - 40.0 eV	M+H

Showing 1 to 4 of 41 entries Previous 1 2 3 4 5 ... 11 Next

The charge filter is not necessary and for *microbeMASST* is, by default, turned off. This will be more relevant for future proteomics applications as we are building related tools with similar interfaces for all the 14,000 proteomics datasets deposited in *MassIVE*.

Please let us know if any additional clarification in the main text is required.

5) When there is more than one metabolome curated for a given microbe, I see that there are available samples and matching samples listed. If the MS/MS spectrum that one enters matches to 50% of that microbe (represented as the pie chart), **is that a complete match to half of the available samples? Or are X number of peaks matched similarly in that half of the samples.** This could be the explanation for the taxa matches tab but it isn't clear – could it also be explained what all the outputs mean on this tab as well?

Thanks for pointing this out as it shows that we need to be clearer about this. In short, it is the number of matching samples, within the user selected search criteria, and not the number of matching peaks. In response, we have now changed lines 192-195 to clarify this. When the users search for an MS/MS spectrum and the parameters used for the search are satisfied (cosine similarity, mass error, number of matching peaks) for a given sample in the *microbeMASST* repository, that is considered a match. For example, let's say we have 10 samples acquired from *Bifidobacterium breve* cultures in the *microbeMASST* repository. The number of deposited samples is then 10. When a MS/MS search is

18performed and a match is found, let's say that only one samples of B. breve presented the matching MS/MS spectrum that the user was searching, the number of found MS/MS matches will be 1 and the proportion of found matches will be 1/10, which is also visualized through the pie chart. If 5 samples of B. breve presented the MS/MS spectrum that the user was searching, the number of found MS/MS matches will be 5 and the proportion of found matches will be 5/10. The 'Taxa matches' tab basically reports the same information that is visualized in the interactive taxonomic tree but as a table that can be filtered and downloaded for third party analysis. Let's say you have found exclusively 5 matches to B. breve, the table will then report 5 matches to B. breve and its proportion (5/10, since we have 10 B. breve samples). Since the matching information is propagated upstream through the lineage of B. breve, it will also be reported 5 matches to the Bifidobacterium genus. Let's say we have a total of 100 samples acquired from bifidobacteria cultures (B. breve, B. bifidum, B. adolescentis, and so on) in the microbeMASST reference database, then the proportion of matches for the Bifidobacterium genus will be 5/100. The same is calculated for each taxonomic rank till the Domain rank.

In the main section we have now written (lines 192-195):

.. number of sample data files containing a match to the queried spectrum within the user selected search criteria, and proportion between the number of sample data files matching the queried spectrum and the number of total available sample data files for that specific taxon in the reference database of microbeMASST.

Due to space constraints, we kept it brief but please let us know if additional clarity is needed.

6) It would be good to mention that the **matches are to partly or fully unidentified metabolites.**

Thanks for the suggestion.

In the main section we have now written (lines 211-213):

It is important to point out that microbeMASST allows to link both partly annotated, through MS/MS match to a reference library spectrum, and fully uncharacterized spectra to possible microbial producers

...

7) In general the **font sizes are too small** on the figures when reading the paper at 100% zoom.

Thanks for pointing this out. The font sizes of the figures have now been increased.

Decision Letter, first revision:

Message: Our ref: NMICROBIOL-23071831A

8th November 2023

Dear Pieter,

Thank you for submitting your revised manuscript "A Taxonomically-informed Mass Spectrometry Search Tool for Microbial Metabolomics Data" (NMICROBIOL-23071831A). It has now been seen by the original referees and their comments are below. The reviewers find that the paper has improved in revision, and therefore we'll be happy in principle to publish it in Nature Microbiology, pending minor revisions to satisfy the referees' final requests and to comply with our editorial and formatting guidelines.

Thank you again for your interest in Nature Microbiology Please do not hesitate to contact me if you have any questions.

Sincerely,

20Reviewer #1 (Remarks to the Author):

The authors have satisfactorily addressed the comments.

Reviewer #2 (Remarks to the Author):

The authors have adequately address this reviewer's comments. No further comments.

Reviewer #3 (Remarks to the Author):

The authors have done a great job responding to the all the comments thoroughly, there are no more comments from me.

Final Decision Letter:

Message: 29th November 2023

Dear Pieter,

I am pleased to accept your Brief Communication "MicrobeMASST: A Taxonomically-informed Mass Spectrometry Search Tool for Microbial Metabolomics Data" for publication in Nature Microbiology. Thank you for having chosen to submit your work to us and many congratulations.

Due to the importance of these deadlines, we ask you please us know now whether you will be difficult to contact over the next month. If this is the case, we ask you provide us with

21the contact information (email, phone and fax) of someone who will be able to check the proofs on your behalf, and who will be available to address any last-minute problems.

Acceptance of your manuscript is conditional on all authors' agreement with our publication policies (see <https://www.nature.com/nmicrobiol/editorial-policies>). In particular your manuscript must not be published elsewhere and there must be no announcement of the work to any media outlet until the publication date (the day on which it is uploaded onto our website).

Please note that *Nature Microbiology* is a Transformative Journal (TJ). Authors may publish their research with us through the traditional subscription access route or make their paper immediately open access through payment of an article-processing charge (APC). Authors will not be required to make a final decision about access to their article until it has been accepted. [Find out more about Transformative Journals](https://www.springernature.com/gp/open-research/transformative-journals)

Authors may need to take specific actions to achieve [compliance with funder and institutional open access mandates](https://www.springernature.com/gp/open-research/funding/policy-compliance-faqs). If your research is supported by a funder that requires immediate open access (e.g. according to [Plan S principles](https://www.springernature.com/gp/open-research/plan-s-compliance)) then you should select the gold OA route, and we will direct you to the compliant route where possible. For authors selecting the subscription publication route, the journal's standard licensing terms will need to be accepted, including [self-archiving policies](https://www.nature.com/nature-portfolio/editorial-policies/self-archiving-and-license-to-publish). Those licensing terms will supersede any other terms that the author or any third party may assert apply to any version of the manuscript.

With kind regards,